# Coherent correlation imaging for resolving fluctuating states of matter

Christopher Klose[1], Felix Büttner[2,3,4 ✉], Wen Hu[3 ✉], Claudio Mazzoli[3,8], Kai Litzius[2], Riccardo Battistelli[4], Sergey Zayko[5], Ivan Lemesh[2], Jason M. Bartell[2], Mantao Huang[2], Christian M. Günther[6], Michael Schneider[1], Andi Barbour[3], Stuart B. Wilkins[3], Geoffrey S. D. Beach[2], Stefan Eisebitt[1,7] & Bastian Pfau[1,8]

Fluctuations and stochastic transitions are ubiquitous in nanometre-scale systems, especially in the presence of disorder. However, their direct observation has so far been impeded by a seemingly fundamental, signal-limited compromise between spatial and temporal resolution. Here we develop coherent correlation imaging (CCI) to overcome this dilemma. Our method begins by classifying recorded camera frames in Fourier space. Contrast and spatial resolution emerge by averaging selectively over same-state frames. Temporal resolution down to the acquisition time of a single frame arises independently from an exceptionally low misclassification rate, which we achieve by combining a correlation-based similarity metric[1,2] with a modified, iterative hierarchical clustering algorithm[3,4]. We apply CCI to study previously inaccessible magnetic fluctuations in a highly degenerate magnetic stripe domain state with nanometre-scale resolution. We uncover an intricate network of transitions between more than 30 discrete states. Our spatiotemporal data enable us to reconstruct the pinning energy landscape and to thereby explain the dynamics observed on a microscopic level. CCI massively expands the potential of emerging high-coherence X-ray sources and paves the way for addressing large fundamental questions such as the contribution of pinning[5–8] and topology[9–12] in phase transitions and the role of spin and charge order fluctuations in high-temperature superconductivity[13,14].

The difficulty of imaging stochastic processes traces back to a conceptual dilemma: to gain spatiotemporal resolution, both full-field and scanning imaging approaches must distribute the detected signal over thousands of pixels. Thus, the better the spatial resolution targeted the larger the signal needed. But the number of sample–probe interactions per volume and time is limited—not only owing to source, optics and detector constraints, but ultimately owing to sample perturbations such as heating, strain, electronic excitations, contrast bleaching, and even sample destruction[15,16]. High spatial resolution therefore requires extensive temporal signal averaging. If, by lack of better knowledge, this averaging is indiscriminate, it leads to a loss of temporal resolution and to motion-blurred images. Under certain conditions, one may recover characteristic spatiotemporal 'modes' of a dynamic system beyond this conventional temporal resolution limit. However, mode decomposition only enhances the signal-to-noise ratio of modes that repeat in time[17,18]; denoising of irregular temporal signals is beyond its scope[18]. Alternatively, mixed-state ptychography may be used to reconstruct static images of the most visited states within the averaging time period[19]. In any case, if the actual sequence of events is of interest, the trade-off between spatial and temporal resolution appears to be fundamental.

CCI overcomes this limit. The key idea is to record snapshots of Fourier-space coherent scattering patterns as raw data, and to exploit that even at low photon count—where imaging is not possible—each scattering pattern contains speckle fingerprints of the real-space state of the system. By combining developments in photon-correlation spectroscopy, nanoparticle tomography[1,2] and genome research[3], we use this sensitivity to accurately classify the state of each snapshot, and hence the timestamps of each state, in a sequence of thousands of frames. Spatial resolution arises independently from the informed average of same-state scattering patterns, which we here convert to real-space images by holographically aided phase retrieval (see Methods). CCI allows us to discover rich fluctuation dynamics in an otherwise well-explored magnetic material, which illustrates the breadth of unexpected physics hidden in fluctuating states of matter and highlights the power of CCI in exploring this territory.

To develop and demonstrate CCI, we study a textbook example of a fluctuating state of matter[20,21]: the finite temperature dynamics of a highly degenerate magnetic ground state. We focus on meandering domains[22] in a $[Pt (2.7 nm)/Co_{60}Fe_{20}B_{20} (0.8 nm)/MgO (1.5 nm)]_{\times 15}$ multilayer[23]—a low-pinning, thin-film perpendicular-magnetic ferromagnet, which represents a class of materials widely considered as a platform for future information technologies[24]. Because of the low pinning in the material, thermal activation is expected to drive magnetic configuration transitions in this system, which is not only critical for any information encoded but also fundamentally a largely unexplored field.

[1]Max Born Institute, Berlin, Germany. [2]Department of Materials Science and Engineering, Massachusetts Institute of Technology, Cambridge, MA, USA. [3]National Synchrotron Light Source II, Brookhaven National Laboratory, Upton, NY, USA. [4]Helmholtz-Zentrum für Materialien und Energie, Berlin, Germany. [5]IV Physical Institute, University of Göttingen, Göttingen, Germany. [6]Zentraleinrichtung Elektronenmikroskopie (ZELMI), Technische Universität Berlin, Berlin, Germany. [7]Institut für Optik und Atomare Physik, Technische Universität Berlin, Berlin, Germany. [8]These authors contributed equally: Claudio Mazzoli, Bastian Pfau. ✉e-mail: felix.buettner@helmholtz-berlin.de; wenhu@bnl.gov

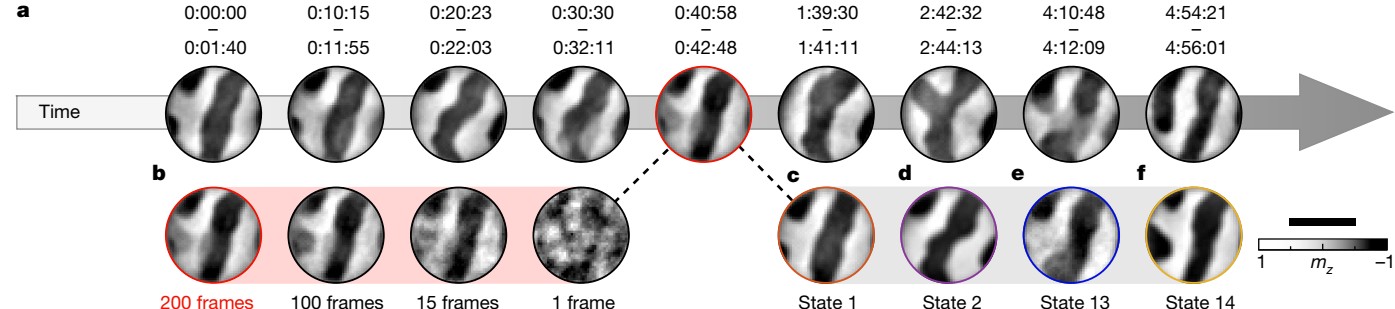

**Fig. 1 | Conventional imaging of fluctuating magnetic domains. a**, Example images of the thermally evolving magnetic maze domain states at 310 K. Each image has been reconstructed by Fourier-transform holography from scattering data sequentially averaged over the indicated time intervals. High-contrast regions indicate stationary behaviour of the local out-of-plane magnetization $m_z$, and grey features correspond to unresolved dynamics during the averaging period. **b**, Example images reconstructed from averages of fewer frames. **c–f**, Disentangled magnetic domain states corresponding to the time interval of **b**. The domain states were identified and reconstructed using CCI. Scale bar, 500 nm.

Here, we investigate the nature of such magnetic configuration fluctuations by recording a nearly continuous sequence of snapshots of the out-of-plane magnetization component $m_z(\mathbf{x}, t)$ in a fixed field of view and at a constant cryostat temperature of 310 K ($\mathbf{x}$ is the spatial coordinate and $t$ is the time interval of the snapshot). Snapshots $I(\mathbf{q}, t)$ were recorded in momentum space ($\mathbf{q}$ is the photon momentum transfer) by resonant X-ray scattering in forward geometry combined with a holographic mask to robustly reconstruct real-space images of $m_z(\mathbf{x}, t)$ by phase retrieval. For technical details, see Methods.

Figure 1a shows images of the natural evolution of $m_z(\mathbf{x}, t)$ reconstructed from successive averages of 200 snapshots (camera frames). Most images show almost binary black-and-white contrast, and the resulting patterns can be interpreted as static maze-like domains of $m_z$, interrupted only by the expected domain wall regions of vanishing contrast. However, as time progresses, these domain states change. Their thermal dynamics is mostly characterized by a gradual evolution of the domain wall network. Surprisingly, the dynamics is discrete (the magnetic states $\{m_z^i(\mathbf{x}), i = 1, 2, 3, \ldots\}$ are countable) and recurring (the same states are revisited after a random set of intermediate states).

Some images contain regions of grey contrast, which suggests that the image is a superposition of dynamically evolving domain states. One can attempt to resolve the intermediate dynamics by averaging fewer exposures (Fig. 1b). However, this comes at a loss of image resolution and contrast. Indeed, we find that without increasing the incident intensity to a perturbative regime, it is impossible to temporally resolve the superimposed magnetic states (Fig. 1c–f) forming Fig. 1b in the traditional way of blindly averaging successive frames.

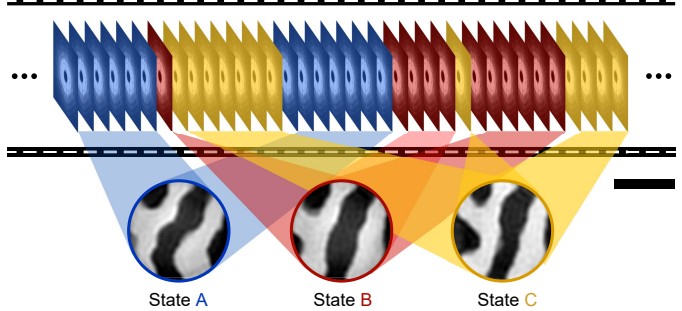

**Fig. 2 | Principle of time-resolved coherent correlation imaging.** Top, sequence of camera frames showing Fourier-space coherent scattering patterns. Coherent correlation imaging classifies scattering frames by their underlying domain state, as indicated by the colours. Bottom, real-space images reconstructed from an informed average of same-state frames. Scale bar, 500 nm.

CCI can solve this dilemma, as sketched in Fig. 2, by classifying all scattering patterns prior to any spatial or temporal averaging by their underlying domain states. High-resolution real-space images of the pure domain states $m_z^i(\mathbf{x})$ can then be reconstructed from an informed signal average $\langle I(\mathbf{q}, t)\rangle_{t | m_z(\mathbf{x}, t) = m_z^i(\mathbf{x})}$. By contrast, temporal resolution emerges from the precision by which each snapshot, and hence timestamp, is assigned to the correct state. The temporal resolution of CCI is hence independent of the spatial resolution, that is, the captured $q$ range, but instead depends on the similarity between states, the speckle contrast, the exposure time, and on the precision (jitter) of the recorded timestamps.

The classification of snapshots is based on a low-noise measure of similarity, or 'distance', between snapshots and is performed in three phases, as shown in Fig. 3. First, based on clustering[4] known from gene identification[3], we obtain an as-complete-as-possible set of magnetic domain configurations encoded in the ensemble of all snapshots. Second, we evaluate the misclassification error of assigning single frames to these domain configurations as a function of the similarity between the configurations. And third, based on an acceptable misclassification error threshold defined by the physics of interest (here we use an error of less than 1%), we group similar domain configurations until the assignment error between frames and groups is below this acceptable threshold. We call the groups 'states' because we can resolve them in space and time, and the configurations within the groups 'internal modes' to emphasize that they are resolved in space but not in time. For details, see Methods and Extended Data Figs. 2–6.

Overall, CCI assigns 99.5% of our 28,800 frames to one of 32 temporally resolved states (Fig. 4a and supplementary video 1), which are composed of 72 internal modes (Extended Data Figs. 4, 6, and supplementary video 2). The dynamics of these states is surprising in several aspects, as summarized in Fig. 4. Already their temporal sequence, depicted in Fig. 4b, reveals unexpected behaviour. First, unlike previously reported thermally activated processes in magnetic materials (such as two-state hopping[20,21,25,26], domain wall creep motion[27], Barkhausen reversal[20,28], and Brownian motion of skyrmions[29]), the dynamics observed is neither cyclic nor evolving, at least not at the timescales resolved here. Second, the number of states is slowly growing, but the dynamics is largely confined to the already explored set of states. And third, we find no obvious temporal order, such as repeating sequences or a preferred switching frequency. Instead, rapid oscillations often randomly follow extended calm periods, and a given state (for example, state 2) can sometimes be stable for tens of minutes and at other times perform excessive hopping on subsecond timescales, see Fig. 4b,d. Assuming that the existence of internal modes does not interfere with the concept of 'lifetime' of a state as a whole, this behaviour is incompatible with the established picture of Arrhenius activation over a fixed energy barrier, where longer lifetimes are expected to be

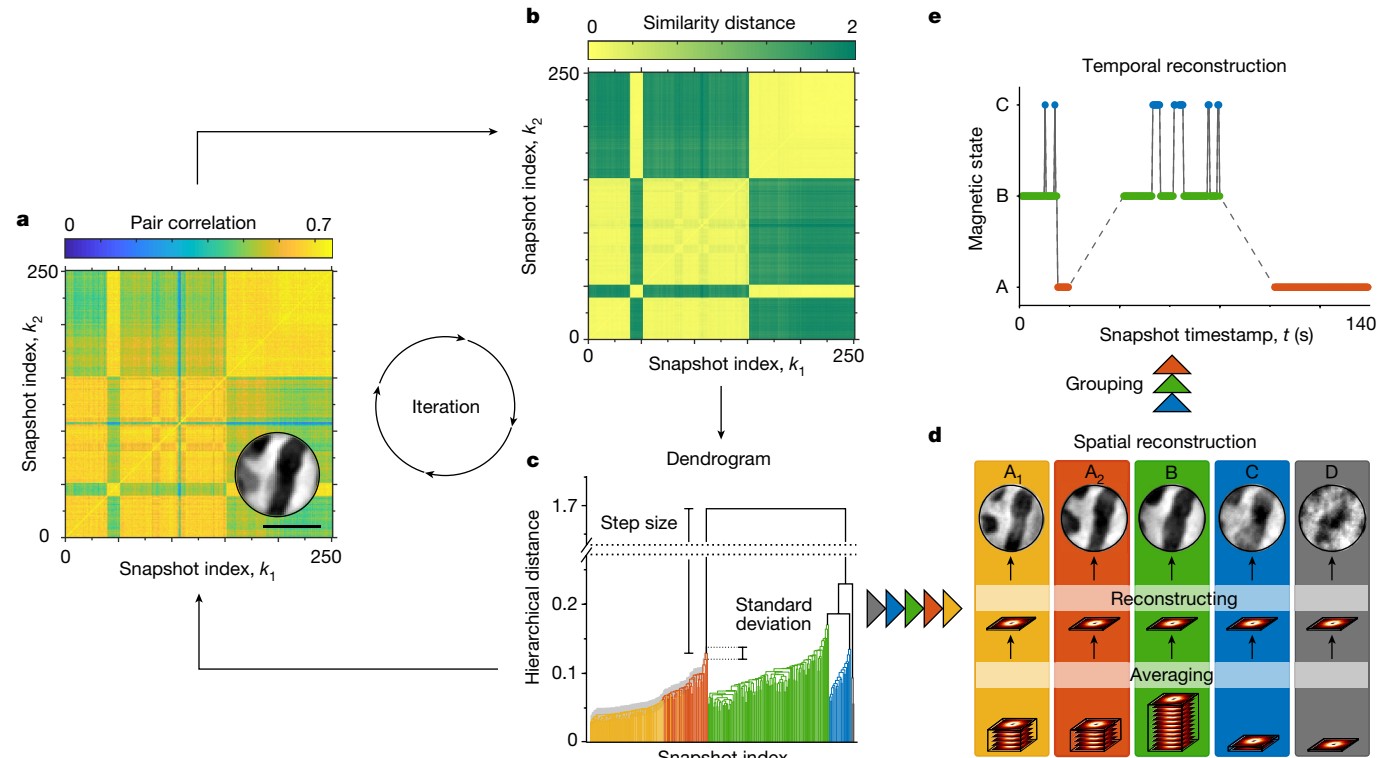

**Fig. 3 | Illustration of the CCI algorithm. a**, Pair correlation map of an exemplary set of 250 camera snapshots. The inset shows the magnetic image reconstructed from the sequential average of all 250 snapshots. **b**, Similarity distance derived from the normalized pair correlation of the pattern in **a**. **c**, Dendrogram calculated from **b** by an agglomerative, hierarchical clustering algorithm. Natural divisions are identified by exceptionally large inconsistencies between the vertical step size and the standard deviation of lower-level links, as indicated by the colour. The complete separation into the clusters in **d** is performed iteratively as indicated. See Methods for details. **d**, Spatially resolved set of domain configurations reconstructed from an informed average of all snapshots of each cluster. **e**, Reconstructed temporal evolution of states, as obtained by grouping domain configurations from **d**, that is, by grouping the internal modes $A_1$ and $A_2$. Dashed lines indicate times without data points. Scale bars, 500 nm.

exponentially suppressed. Instead, this observation suggests a dynamic energy landscape due to correlation effects of the multi-domain state.

A more detailed picture of the dynamics emerges when considering the similarity distance of transitions, as shown in the transition network in Fig. 4c. The extent of the network in this similarity space remains almost constant after state 7 (states are numbered by occurrence), which underlines the non-evolving character of the dynamics. More importantly, however, the hopping frequency is not obviously linked to the hopping distance (compare, for example, states 2, 3, 4, 27). Instead, an underlying order arises if we combine the distance and the frequency of transitions observed (see Methods). With this metric, the network separates into agglomerates, see Fig. 4b,c. Transitions within these agglomerates are much more common than between them. Interestingly, many agglomerates are centred around one or a few 'linker states' that mediate most of the intra- and inter-agglomerate transitions, for example, states 1, 2 and 7.

Although the time and spatial frequency domain data are sufficient to extract many relevant features of the dynamics (Fig. 4b,c), spatial and spatiotemporal imaging is key to understanding the microscopic mechanism underlying these observations. Figure 4e illustrates the level of detail that is accessible from real-space data. First, from the spatially resolved internal modes, we can reconstruct a map of possible domain-wall pathways in the field of view (Fig. 4e and Extended Data Fig. 7c). These paths fan out in some parts of the sample whereas in others their distribution collapses to a single point—a behaviour reminiscent of a snaring guitar string. Following this analogy, we identify the points where many domain-wall paths merge as attractive pinning sites (see Methods and Extended Data Fig. 7d for the exact criteria). The distribution of attractive pinning sites is shown in Fig. 4e.

In addition to attractive pinning sites, our data also point to the presence of repulsive interactions that restrict the positions of domain walls. Specifically, we find extended areas where no quasistatic domain wall configurations were observed during the acquisition. Often, even straight-line connections between attractive pinning sites are missing. From the spatiotemporal data we can extract how often $m_z$ has switched in these areas (see Extended Data Fig. 7e) and hence estimate how easily they can be traversed by a domain wall. Combining this information, we have identified the strongest repulsive pinning sites, as shown in Fig. 4e.

We are now in a position to explain the physics behind the dynamics observed. At the core of it is the conceptual difference between attractive and repulsive pinning sites that derives from a strong topological constraint: domain walls cannot end in the interior of a material. As illustrated in Extended Data Fig. 8, this constraint implies that for domain walls, attractive pinning sites have a point-like action (domain walls are pinned most strongly to the bottom of a depression in the energy landscape), whereas repulsive pinning sites have a two-dimensional character (energetically elevated areas are avoided entirely). This difference explains the separation of the dynamics into agglomerates. Specifically, inter-agglomerate transitions are predominately associated with hopping over one or more repulsive pinning sites, whereas most intra-agglomerate transitions involve only attractive pinning sites (see Extended Data Fig. 9, where we also show exceptions from this rule). Owing to their area-shaped action, repulsive pinning sites add an energetic cost to an entire domain wall section during a transition. Likewise, they inhibit 'zipper-type transitions', that is, one-by-one reordering of a domain wall between multiple attractive pinning sites. These conceptual differences lead to lower activation energies of intra-agglomerate transitions and therefore explain why they occur more frequently than

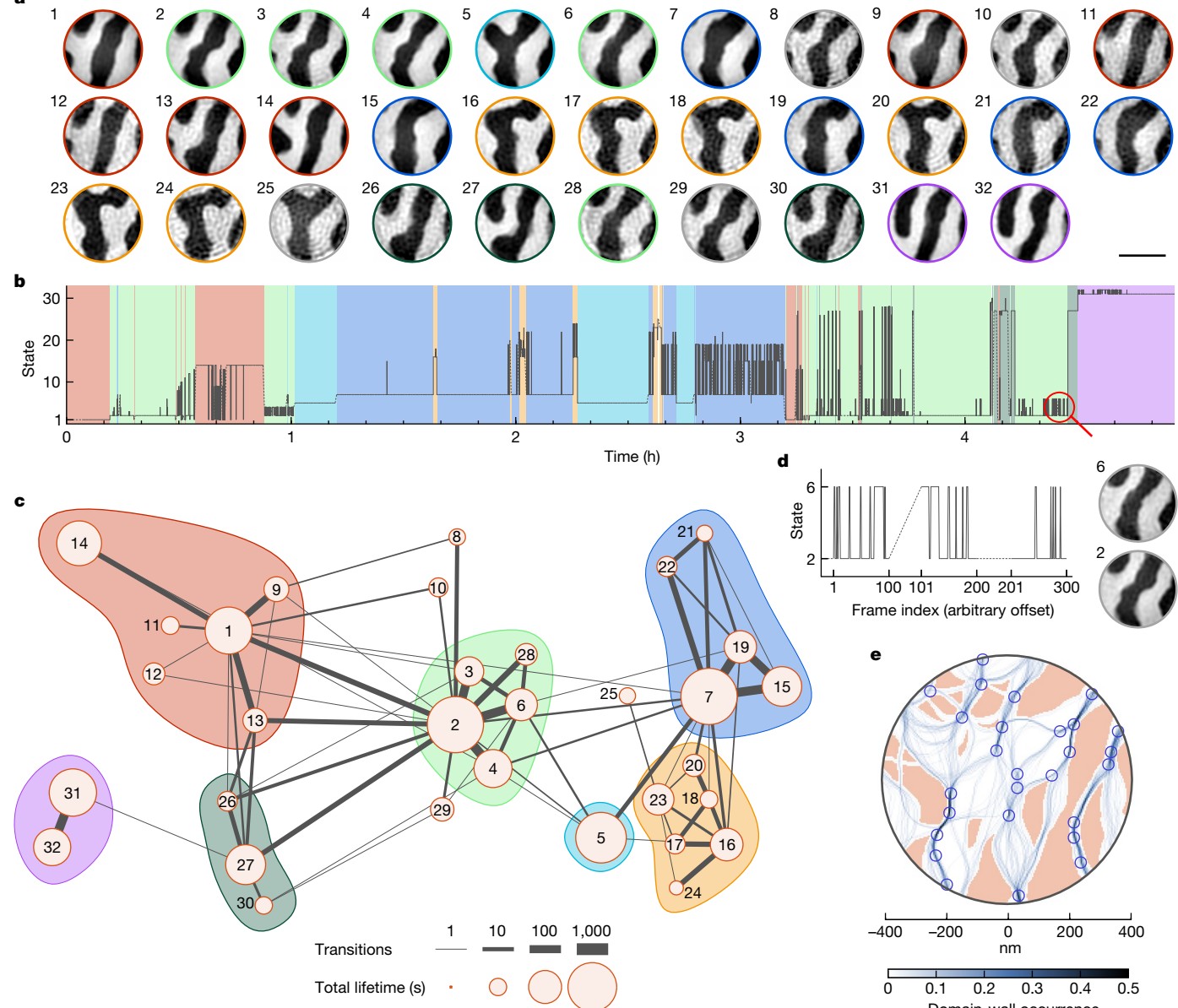

**Fig. 4 | CCI resolved dynamics of the fluctuating domain network. a**, Real-space images of all domain states. The circular coloured borders refer to the agglomerates indicated in **c**. **b**, Temporal evolution of the states. The background shades refer to the agglomerates. Dashed lines indicate times without data points. **c**, Transition network of the domain configurations. The distance between nodes approximately represents the Pearson distance of the corresponding states. The width of the connecting lines indicates the number of observed pair transitions. The diameter of the circular orange contours illustrates the total observed lifetime of each state. Background colours highlight state agglomerates. **d**, Magnification of the region marked with a magnifying glass in **b**. The displayed real-space images were reconstructed exclusively from the corresponding set of frames. **e**, Map of attractive (blue dots) and repulsive (red areas) pinning sites. The background shows the position of the domain walls and their relative occurrence in the internal modes. Scale bar, 500 nm.

inter-agglomerate ones. Moreover, the fragmentation of the material by extended prohibited areas explains why we observe such a small number of states out of all the possibilities to connect the attractive pinning sites. Finally, the seemingly chaotic time trace can be attributed to the intrinsic long-range stray-field interactions, which add a nonlocal, configuration-dependent contribution from areas outside the field of view to the energy landscape.

Stochastic dynamics are very common in systems with spatially varying order parameters. Examples include domain fluctuations in antiferromagnets[30,31], fluctuations of spin and charge density waves in high-transition-temperature ($T_c$) superconductor materials[13,14], the dynamics of frustrated systems such as artificial spin-ice materials[32,33],

thermally activated motion of crystallographic domain boundaries[34], as well as virtually any dynamics during or close to a phase transition. These collective dynamics at the nanometre scale provide, on one hand, an exciting playground in which to explore the underlying—often competing—fundamental interactions, and, on the other hand, to exploit the richness of the emerging physics for advanced devices. Many of such phenomena were previously studied by coherent X-ray or electron scattering experiments, but, with a few exceptions, not via high-resolution real-space imaging owing to insufficient signal on a timescale faster than the relevant dynamics or the state lifetimes. Hence, many details uniquely accessible by time-resolved imaging—such as the shape, morphology, connectivity and topology of locally

ordered regions, the dynamics of these properties, and their relation to material inhomogeneities—have remained unexplored, preventing a fundamental understanding of the underlying materials physics. CCI now enables such direct imaging research, at least for all those systems which are currently investigated in reciprocal space and which return, within the duration of the experiment and within a limited field of view, to the same configuration or a close approximation thereof (such that 'binning' into a discrete set of 'states' is possible).

CCI is a natural extension of any reciprocal-space coherent imaging technique and may be combined with mode decomposition techniques to analyse the resulting real-space dynamics[18] or to eliminate timing uncertainties of the nominal time stamps of the reconstructed images[17]. CCI is applicable to fully or partially coherent sources from visible light to soft or hard X-rays or even electrons, both in transmission and reflection geometries[35], and requires no modification of existing coherent imaging set-ups. Moreover, CCI promises a change of paradigm in time-resolved imaging at sources such as free-electron X-ray lasers and high-harmonic generation sources, where a single probe pulse may carry enough information to identify a state, in particular when combined with few-photon correlation techniques[2] and machine-learning classification[36]. Under such conditions, CCI overcomes the triggerability and repeatability limitations of pump–probe schemes and enables nonperturbative imaging of both equilibrium and nonequilibrium dynamics. In conjunction with the next generation of free-electron X-ray lasers running at MHz repetition rates, real-time imaging with submicrosecond temporal resolution as well as nondestructive pump–probe imaging with femtosecond resolution and wavelength-limited spatial resolution will be feasible. This advancement is illustrated in Extended Data Table 1, where CCI is compared to existing coherent imaging techniques.

Our results demonstrate that coherent scattering signals encode enough information to overcome the long-standing dilemma between spatial and temporal resolution. By means of CCI, the spatiotemporal evolution of a system is now accessible from such coherent scattering data down to the single-frame acquisition limit, even in the absence of repeating characteristic sequences. We have demonstrated that this approach performs well even in weak-contrast systems where differential imaging is required. CCI revealed otherwise inaccessible insights into the domain-wall dynamics of degenerate maze domain states, where it uncovered an intricate transition network with unexpected agglomeration in similarity distance space and a seemingly chaotic temporal behaviour. Most importantly, CCI provided spatiotemporal data based on which we developed a general mechanistic understanding of how these complex dynamics arise from a combination of attractive and repulsive pinning and intrinsic configurational energies, with direct consequences for any potential devices using domain walls. With these capabilities, CCI promises to become a powerful tool to answer some of the large open questions in condensed-matter physics and materials science, such as the role of thermal fluctuations and material defects in microelectronics[37] or batteries[38], the role of fluctuations and nanometre-scale material inhomogeneities in high-$T_c$ superconductivity[6,7,13,14], the dynamics of magnetic monopole excitations in frustrated spin-ice systems[32,33], and the role of fluctuations and topological states in ultrafast phase transitions[5,10–12,39,40].

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

## Methods

### Sample fabrication

The thin-film magnetic multilayer [Pt (2.7 nm)/Co$_{60}$Fe$_{20}$B$_{20}$ (0.8 nm)/MgO (1.5 nm)]$_{x15}$ with Pt (2.7 nm) capping layer on a Ta (2.3 nm)/Pt (3.7 nm) seed layer was sputter-deposited onto a silicon nitride membrane. Ta, Pt and Co$_{60}$Fe$_{20}$B$_{20}$ were grown by d.c. magnetron sputtering and MgO was grown by radio frequency sputtering at Ar pressures of 3.5 mtorr (for Pt) and 3 mtorr for all other materials. The material has a Curie temperature of 650 K, as reported previously[23]. The zero-temperature, zero-field magnetic ground state of this material is a state of parallel stripe-shaped domains of the out-of-plane magnetic order parameter $m_z$, but any meandering configuration of these stripe domains is almost equal in energy and in practice all these configurations are energetically degenerate[23].

The back side of the silicon nitride membrane was coated with [Cr (5 nm)/Au(55 nm)]$_{x20}$ to block the incident soft X-rays[41]. Three circular apertures were milled by focused-ion-beam lithography into this opaque layer, one with 720 nm in diameter defining the field of view and two reference holes 60 nm and 40 nm in diameter providing reference beams for holography. These hole diameters represent a geometry that was developed for time-resolved X-ray holography of out-of-plane magnetic domains[41–43]. They represent a good compromise between sensitivity for small changes (which is easier revealed with smaller object holes) and contextual information (for which larger field of views are preferred).

### Data acquisition and sample environment

The experiment was performed at the coherent soft X-ray scattering beamline (CSX) of the National Synchrotron Light Source II, Brookhaven National Laboratory. Partially coherent, circularly polarized X-rays were focused onto the sample in transmission geometry. A 50-μm upstream pinhole was used to increase the coherence and to reduce the overall flux to limit beam-induced heating. Magnetic contrast was achieved by tuning the X-ray energy to the Co L$_3$ absorption edge (778 eV), where the X-ray magnetic circular dichroism (XMCD) contrast is maximum[44]. A Sydor Fast CCD camera, mounted in-vacuum 33 cm downstream of the sample, recorded the coherently scattered and phase-encoded X-rays as a hologram. The direct beam was blocked by a beamstop. To image the dynamics in a time series, holograms (frames) were recorded continuously and stored individually with their respective timestamps. To separate magnetic from topographic contrast (see below), the helicity was reversed every 100 frames. The exposure time of each single frame was set to 0.3 s, slightly below the saturation limit of individual detector pixels. The effective readout time was 0.087 s. A full set of stacks at both helicities was recorded in ~120 s. This includes the time to change the undulator. Timestamps of the frames were recorded by the beamline controls and acquisition system, with computer precision and hence negligible temporal error compared to the exposure and readout time. That is, the temporal resolution of our experiment was 0.387 s. A liquid helium flow cryostat was used to control of the sample temperature with $\Delta T < 0.01$ K precision.

### Algorithm to subtract static topography holograms

Topography holograms $I_{sum}$ were recorded as a sum of positive and negative helicity holograms. Only holograms without dynamics during the acquisition were used for topography data. That is, we only considered sets where the magnetic reconstruction from the difference of positive and negative helicity holograms showed full contrast across the entire field of view. The topography holograms were manually updated along the temporal frame sequence as needed. In total we used 26 topography holograms, each averaged over at least 100 frames per helicity. To determine if the topography image still represented the current state of the sample, we determined if the topographically induced airy pattern from the circular aperture was successfully suppressed in the difference holograms. Moreover, we checked that our topography holograms were free of any magnetic contrast via the pair correlation maps. To this end, we note that spurious magnetic contribution would produce an offset in the magnetic pair correlation map. The sign of this offset would be inverted at every helicity change, leading to a characteristic checkerboard pattern in the correlation maps. We confirmed that such patterns are absent in our data.

Subsequently, we subtracted $I_{sum}$ from all magnetic holograms $I_\pm$ recorded with either positive or negative helicity X-rays to obtain the difference hologram $I_{diff}$ without mask scattering. To compensate for intensity drift, we used a dynamic factor $\alpha$, that is, $I_{diff} = \pm(I_\pm - \alpha I_{sum})$. The dynamic factor was automatically determined for each image from the scalar product of the full topographic and the magnetic holograms as $\alpha = \langle I_\pm, I_{sum}\rangle/\langle I_{sum}, I_{sum}\rangle$. This equation is based on the approximation that topography and magnetic signal are uncorrelated and hence orthogonal. Importantly, this approach leaves only one time-dependent signal in the reconstruction process ($I_\pm$) and, moreover, produces the same output $I_{diff}$ for the same state regardless of whether it was recorded with positive or negative helicity light. Thus, after this step, the temporal resolution of CCI equals the temporal resolution (here the acquisition time) of the original single frame $I_\pm$.

### Real-space image reconstruction

Images in Figs. 1, 3 were reconstructed by Fourier-transform holography (FTH)[42,45,46], which is a linear operation and therefore best suited to illustrate the effect of limited signal-to-noise ratio (SNR) and blind averaging. The final, high-SNR images of the magnetic states and modes (Figs. 2, 4, Extended Data Figs. 4, 6, 7, 9 and supplementary videos) were reconstructed by holographically aided iterative phase retrieval[47,48], which offers increased resolution and contrast of the resulting images as long as the underlying scattering data are of low noise.

Whereas FTH images were processed based on difference holograms, our phase retrieval algorithm relies on the positive values in the scattering plane. Therefore, we started by calculating the single helicity holograms for every mode discerned by our CCI algorithm using the following equation:

$$I_{\pm,i} = \sum_{k \in \phi^i} I_{\pm,k} = \sum_{k \in \phi^i} \frac{2\alpha_k I_{sum,k} \pm I_{diff,k}}{2} \tag{1}$$

where $I_{\pm,k}$ is a frame belonging to the set of frames $\phi^i$ assigned to a particular mode $i$, $I_{sum,k}$ is the topography image associated with that specific frame and $\alpha_k$, $I_{diff,k}$ are its associated dynamic factor and difference image, respectively (see Methods section 'Algorithm to subtract static topography holograms' for the definition of the dynamic factor, $\alpha$). Such a procedure allowed us to obtain single helicity images for each frame with the photon statistics of both helicities.

The resulting holograms were then centred, and any present (readout) offset was subtracted. A support mask for the phase retrieval algorithm was manually generated by selectively thresholding the absolute value of the FTH reconstruction of the average hologram. The same mask was used to reconstruct all images. The holograms were then fed to an iterative phase retrieval algorithm designed for dichroic imaging[48]. First, the positive helicity exit wave of the sample was reconstructed using a combination of two different phase retrieval algorithms. The starting guess used was derived from the support mask multiplied by a constant, adjusted according to the intensity of the input hologram. We first applied a modified version of the iterative relaxed averaged alternating reflections (RAAR) algorithm[49] (700 iteration steps with a relaxation parameter $\beta$ smoothly going from 1 to 0.5), where the non-negativity reflector constraint was removed, owing to its inadequacy to deal with complex holograms. The positive helicity reconstruction was completed by the error reduction algorithm[50] (50 iteration steps).

Then, the negative helicity exit wave was reconstructed using the error reduction algorithm (50 iteration steps) using the phase of the reconstructed positive helicity complex hologram as a starting guess.

This approach enables us to avoid any shift between the two different helicity images. Finally, the magnetic reconstruction was obtained by computing the ratio between the difference and the sum of the two reconstructed images to extract the pure XMCD contrast.

The phase retrieval algorithm yields reconstructions with higher contrast and better resolution compared to the FTH reconstruction. We achieve diffraction-limited resolution of 18 nm for internal modes with 250 frames or more contributing to the reconstruction. The resolution was determined via Fourier ring correlation[51] using the half-bit threshold criterion[52]. Displaying the ratio between the difference and sum, instead of the bare difference, allows to neglect topographic features and widens the field of view of the technique to 810 nm, as the borders of the object hole partially covered in gold present the same contrast as the centre, despite less light being transmitted.

## Algorithm to calculate magnetic pair correlations in Fourier space

In our X-ray holography geometry, we can determine the magnetic pair correlation directly in Fourier space, as illustrated in Extended Data Fig. 1. We start by subtracting the topography hologram from the recorded single-helicity hologram to obtain the difference hologram $h_{\mathrm{diff}}$ (Extended Data Fig. 1a). Fourier-transforming the difference hologram yields the so-called Patterson map (Extended Data Fig. 1b). The Patterson map includes four reference-reconstructed real-space images of the out-of-plane magnetization $m_z(\mathbf{x}, t)$ in the off-centre region, and the interference between object hole charge scattering and magnetic scattering in the centre.

We next eliminate the reference-induced modulations as these are mostly adding noise, owing to their sensitivity to fluctuations of the incident beam. We therefore first crop the central part of the Patterson map excluding the reconstructions and, second, transform back to Fourier space. The resulting filtered difference hologram $\tilde{I}_{\mathrm{diff}}$, depicted in Extended Data Fig. 1c, exhibits high intensity at small reciprocal vectors $\mathbf{q}$ and needs to be flattened before performing the correlation analysis. Mathematically, $\tilde{I}_{\mathrm{diff}} \propto \mathrm{Re}(\mathcal{F}[m])\mathcal{F}[T_{\mathrm{mask}}]$, where $\mathcal{F}[T_{\mathrm{mask}}]$ is the Airy-disk scattering of the approximately binary transmission $T_{\mathrm{mask}}$ of the circular mask defining the field of view. This identity is derived in ref. [53]. We therefore extract the pure magnetic scattering $S_i = \tilde{I}_{\mathrm{diff}}(t_i)/\sqrt{\tilde{I}_{\mathrm{sum}}} \propto \mathrm{Re}(\mathcal{F}[m])$ by dividing the filtered difference hologram by the square root of the topographic sum hologram $\tilde{I}_{\mathrm{sum}} \propto \mathcal{F}[T_{\mathrm{mask}}]^2$. Then, the Fourier-space pair correlation matrix (Extended Data Fig. 1d), calculated using

$$c_{ij} = \frac{\langle S_i, S_j \rangle}{\|S_i\| \, \|S_j\|}, \tag{2}$$

is essentially equivalent to the pair correlation matrix of the underlying real-space textures $m_z(\mathbf{x}, t)$, see ref. [53]. Note that $\langle ., . \rangle$ is the scalar product of the unravelled pixel arrays and $\|.\|$ is the norm according to this scalar product.

## Robust classification of frames

Our method to classify frames corresponding to the same physical state is illustrated in Fig. 3. It is based on two ingredients: (i) a low-noise metric $d_{ij}$ that quantifies the pair-wise similarity ('distance') of two camera frames $\phi_i$ and $\phi_j$ recorded at times $t_i$ and $t_j$ and (ii) an iterative, hierarchical clustering algorithm[3,4] that groups frames into same-state sets.

We start with the frame-wise pair correlation map of magnetic textures, $c_{ij}$, see previous section. As shown in Fig. 3a and Extended Data Fig. 2, already $c_{ij}$ resolves clear grid-shaped fingerprints of state changes at a single-frame level, with approximately ten times better signal-to-noise ratio than a similar analysis based on holographic real-space reconstructions (see Extended Data Fig. 3). We further reduce the noise of the metric by the following consideration: If $S_\alpha$ and $S_\beta$ represent the same state (A) then the correlation of $S_\alpha$ with any arbitrary frame should be the same as the correlation of $S_\beta$ with that

frame. This follows from the transitivity of the equivalence relation. Hence, each pair correlation $c_{\alpha\gamma}$ for any $\gamma$ is a fingerprint of state A and the entire vector $c(S_\alpha) \equiv (c_{\alpha 1}, c_{\alpha 2}, ..., c_{\alpha n})$ has considerably better photon statistics than the single scattering pattern $S_\alpha$ alone. We therefore define our low-noise fourth-order Pearson distance metric $d_{ij}$ as

$$d_{ij} = 1 - c_{ij}^{(4)} = 1 - \frac{\langle c(S_i) - \bar{c}(S_i), c(S_j) - \bar{c}(S_j) \rangle}{\|c(S_i) - \bar{c}(S_i)\| \, \|c(S_j) - \bar{c}(S_j)\|}, \tag{3}$$

where $\bar{c}(S_i) = \frac{1}{n} \sum_{j=1}^{n} c_{ij}$ is the mean of the vector $c(S_i)$. That is, the matrix $c_{ij}^{(4)}$ is the normalized Pearson correlation matrix of $c_{ij}$. The resulting distance matrix is shown in Fig. 3b.

On the basis of the measure of distance, we identify distinctive clusters of highly similar frames through an iterative variant (Fig. 3a–c) of a concept know from gene identification[3] as UPGMA (unweighted pair group method with arithmetic mean) to construct a hierarchical cluster tree from $d_{ij}$. In this bottom-up process, the nearest two clusters are combined into a higher-level cluster, starting with single frames on the lowest level[3,4] and ending with a single final cluster. The UPGMA clustering was performed using the MATLAB function 'linkage' specified with the 'correlation' metric to compute the proximity of two scattering patterns $S_i$ and $S_j$ by autocorrelation of the input matrix $d_{ij}$. This additional application of the correlation function results in a nonlinear similarity metric enhancing the numerical separation of high correlation values (similar states) and low correlation values (dissimilar states). The resulting so-called dendrogram is presented in Fig. 3c. The vertical level, or height, of each link in the dendrogram corresponds to the distance (abstract, nonlinear in our case) between the linked clusters. If two linked clusters represent the same physical state of the sample, the step size, or height difference, between the old and the new hierarchy level is small. Conversely, a large step, also known as an inconsistency, indicates a natural division in the data set, that is, the merging of different physical states. The largest step size is naturally observed at the topmost node in the tree. We aim to make the most robust classification and therefore split the tree only at the top node into two subclusters. Instead of dividing the dendrogram into a complete set of clusters in a single step, we iterate through the whole clustering procedure, separating the dendrogram in each iteration at most into two subgroups (at the topmost link). The iterations enable us to extract domain configurations beyond the conventional UPGMA, for example, to separate configurations $A_1$ and $A_2$ in Fig. 3c.

The next iteration step is to decide if our two identified subclusters represent pure or mixed states, that is, if they comprise frames from a single state or multiple states. To this end, we quantify the distinctness of the topmost link in the dendrogram by the inconsistency coefficient $\xi$, where $\xi$ is given as the ratio of the average absolute deviation of the step height to the standard deviation of all lower-level links included in the calculation. Both the step height and the standard deviation are indicated in Fig. 3c. On this basis, we identify all clusters that undercut an inconsistency threshold as pure-state clusters whereas, complementarily, mixed states are present in clusters exceeding this threshold.

The iteration steps are repeated for mixed-state clusters until—within our temporal resolution and available signal-to-noise ratio—complete separation into clusters containing only pure states is achieved. The frames of each pure-state cluster are then averaged, and a real-space image of the corresponding domain configuration is reconstructed (Fig. 3d). The knowledge of which frames entered each cluster, and the time stamps of when these frames were recorded, yields the temporal reconstruction, as shown in Fig. 3e (see Methods section 'Estimation of the temporal discrimination threshold and reconstruction of the 32 states' for more details about the temporal reconstruction).

Applied to our data, we find that the modified clustering algorithm performs well for thresholds of the inconsistency coefficient $\xi$ in the range 1.7 to 2.2. The upper limit is determined by the minimal sensitivity needed

to separate the most evident mixed-states (for example, identified by motion-blurred contrast, as in Fig. 1b). The lower limit is set by the total number of frames (here 99%) that can be assigned to clusters that yield acceptable real-space reconstructions, which we assume as 15 frames per cluster. Specifically, we chose $\xi = 1.85$ because it produces the lowest frame misclassification probability while assigning 99.5% of our 28,800 frames to clusters with >15 frames (see Methods section 'Estimation of the temporal discrimination threshold and reconstruction of the 32 states' for more details about the evaluation of the frame misclassification).

We note that major drawbacks of hierarchical clustering are the so-called chaining phenomenon and the sensitivity towards outliers, merging inconsistent clusters or creating minor cluster with few elements. Whereas the former is effectively minimized in the iteration process by choosing a sufficiently low inconsistency threshold, the latter leads to the fragmentation of a pure state into several inseparable clusters. In our case, the algorithm finds 119 magnetic configurations, of which many are visibly indistinguishable. To eliminate duplicates in this set, we calculated the similarity between all identified clusters $A$ and $B$ by averaging the pair correlation values $c_{AB}$ for all combinations of frames $\phi_A$ and $\phi_B$ belonging to these clusters. Using a conventional (noniterative) agglomerative UPGMA algorithm we then processed the resulting reduced 119 × 119 cell cluster-correlation matrix to group clusters that actually represented the same domain configurations. We verified that our final clusters are pure states by applying the same iterative scheme as mentioned above.

The result is a set of 72 domain configurations forming the basis for further separation into discretized internal modes (discretized versions of the just spatially resolved domain configurations) and states (spatiotemporally resolved configurations).

## Discretizing of the internal modes

To extract the positions of domain walls from sometimes noisy or inhomogeneously illuminated real-space images reconstructed by the phase retrieval algorithm, we discretized the 72 domain configurations identified in the cluster analysis (see Extended Data Fig. 4). To this end, we started with a low-pass filter: the cluster-averaged difference holograms were first cropped with a circular binary mask before reconstructing real-space images to reduce high-frequency noise in the images. The radius of the filter in Fourier space was determined individually for each internal mode as the maximum $q$ beyond which the noise dominates over the total magnetic scattering signal of all frames of that mode. Although this filter reduces the spatial resolution, the location of the domain wall remains unaffected. The domain segmentation was based on the sign of the reconstructed phase $\varphi(r)$ of the complex-valued images instead of its real part because the phase shows the most distinct separation of the up and down magnetization. The reconstructed phase was centred around zero to achieve maximum contrast observed by a phase transition of $\pm\pi$ at the domain wall locations. We considered domains smaller than 50 pixels as artefacts caused by noise and merged them therefore with their surrounding domains. Images that were strongly affected by the artefacts were adjusted manually.

Some of the original real-space reconstructions of the 72 internal modes show evidence of mixed-state superposition, see Extended Data Fig. 4b. Such mixed states inevitably emerge if the domain state changes during the acquisition of a single frame (that is, if the dynamics is faster than our minimum temporal resolution). Hence, simply binarizing a domain image as the one shown in Extended Data Fig. 4b would constitute a loss of information. However, we find that the whole set of binarized internal modes forms a representative basis, based on which the original greyscale images can be decomposed, as illustrated in Extended Data Fig. 4b. In practice, we evaluated the similarity between the domain images and all binary internal modes by the real-space correlation[53]. In a next step, we applied a multilinear regression to decompose each individual greyscale image into those binary internal modes which have >88% similarity to the greyscale image.

We find that the weighted superposition of internal modes accurately represents all domain configurations, with the exception of state 32, where an additional binary domain configuration 73 was manually created (we attribute this to the fact that state 32 is the last state in our time series and insufficient data were available to automatically decompose it). The discrete representation of all 72 internal domain modes is shown in Extended Data Fig. 6. The set of original domain images along with their low-pass filtered phase images, their adjusted binarized versions, and their decomposition into binary internal modes is compiled in supplementary video 2.

## Estimation of the temporal discrimination threshold and reconstruction of the 32 states

Cluster analysis is a method to classify data into natural divisions with similar features, for example, coincident magnetic domain states in this work. Any such algorithm applied to noisy data will eventually produce assignment errors. These errors have a crucial role in the temporal reconstruction, where every misclassified frame is a falsely detected or neglected transition. However, finding the genuine assignment becomes increasingly difficult for states that are very similar (see domain configurations $A_1$ and $A_2$ in Fig. 3d). Of the 72 internal modes identified by the clustering algorithm, some exhibit real-space similarities of up to 99%. Given this extremely close similarity, the question arises how accurate the assignment of a single frame to a particular magnetic state is. We address this question by estimating the frame misclassification probability, which we derive from the robustness of assigning a frame between two clusters.

In the bottom-up clustering approach used in our analysis, links between clusters created on lower levels strongly influence higher-level clustering. The whole linkage tree, that is, the dendrogram, is particularly sensitive to the links created at lowest levels for low-distance high-noise data, where small changes of the initial distance may lead to deviating final assignments. Conversely, the robustness against variations of the initial set of frames is a measure of the quality of a clustering assignment. Based on this rationale, we developed the following analysis to verify the accuracy and sensitivity of our cluster analysis.

In our analysis, we validate the clustering assignment of frames between every pair of clusters $A$ and $B$. We assume that our full cluster analysis already outputs a representative estimate of the members $\phi^A$ and $\phi^B$ belonging to $A$ and $B$, respectively. For error analysis, we now compare clustering runs starting with different initial frame sets—on the one hand, the original frame set $\phi^A + \phi^B$ and, on the other hand, ten newly assembled subsets $\widetilde{\phi}^A + \widetilde{\phi}^B$. Each subset $\widetilde{\phi}^A$ and $\widetilde{\phi}^B$ contains a randomly selected half of the parent frame set $\phi^A$ and $\phi^B$, respectively. We separate the assembled frame sets into exactly two clusters using the same iterative agglomerative UPGMA algorithm as described previously. The robustness of the clustering is evaluated by comparing the original frame-to-cluster assignment with the new assignment, that is, if frames that are grouped in $A$ or $B$ are still grouped in $\widetilde{A}$ or $\widetilde{B}$, respectively. Otherwise, frames are considered as misclassified and we define the misclassification probability as the average fraction of misclassified frames in $\widetilde{\phi}^A + \widetilde{\phi}^B$.

For the majority of mode pairs (87%) we find perfect coincidence in the clustering with random subsets. We find that the robustness of the frame classification substantially drops if at least one of the clusters contains a very low number of frames. We exclude clusters with less than 100 frames—the number of frames for a standard FTH image of a single hologram stack—from further analysis because recording more frames is not a fundamental challenge.

In Extended Data Fig. 5, we show the frame misclassification probability of all remaining mode pairs as a function of their similarity, represented by their pair correlation. Notable misclassification rates occur for very similar domain configurations only. Specifically, we find that the frame misclassification probability is below 1% for similarities of the underlying magnetic textures of up to 93.8%. We define this as

the sensitivity threshold in our analysis. Internal domain modes with higher similarity were grouped allowing their temporal discrimination with an acceptable misclassification rate, as shown in Fig. 3e. These groups of modes are referred to as spatially and temporally resolved magnetic 'states'. In practice, we successively combined the nearest modes until the distance between any pair of modes of two dissimilar states exceeded the 100% − 93.8 % = 6.2% similarity distance defined by the sensitivity threshold. An overview of the identified states and their associated internal domain modes is shown in Extended Data Fig. 6. We note in this context that the correlation between domain wall hopping in different parts of the field of view is not an artefact of the clustering algorithm but instead emerges from the intrinsic physics of the material, namely from the desire of the material to retain a constant domain width as set by the competition of domain wall energies and long-range stray field energies.

The number of states and internal modes may vary with particular choices of model parameters, but the conclusions drawn in the text remain robust in a broad range of parameters discussed above. We note further that this approach of 'binning' of similar configurations into 'states' makes CCI broadly applicable even to systems exhibiting nonrecurring dynamics, especially when combined with a fixed and finite field of view.

### Determination of agglomerates in the transition network

Already in its raw representation, the transition network (Fig. 4c) visually segregates into agglomerates of states. To quantify this observation, we defined a new distance metric that takes into account the similarity distance as well as the frequency of transitions between states. To this end, we use the simplest approach, where we just add the two distance measures as

$$d_{AB} = 2 - \left( c_{\text{state}}(A, B) + \frac{\widetilde{\theta}(A, B)}{\widetilde{\theta}_{\text{max}}} \right).$$

where $d_{AB}$ is the distance between states $A$ and $B$ according to this new metric, $c_{\text{state}}$ is the state similarity distance and $\widetilde{\theta}(A, B) = \log(1 + \theta T)$ is the logarithm of the transition frequency $\theta$ between $A$ and $B$, $T$ is the duration of the experiment, and $\widetilde{\theta}_{\text{max}}$ denotes the highest observed $\widetilde{\theta}$ across all states. We then applied a UPGMA hierarchical clustering algorithm to create a dendrogram from the states according to this distance metric (Extended Data Fig. 10). The separation threshold that defines the agglomerates was manually set at the large step around 0.38 in this dendrogram.

### Determination of attractive pinning sites

Our approach to determine the position of attractive pinning sites is based on the domain-wall occupation probability and the averaged domain-wall curvature. First, we extracted domain-wall locations that were observed in at least 20% of all internal modes shown in Extended Data Fig. 7c. Among these hotspots, there are point-like and line-shaped areas with an increased occupation probability indicating pinning sites. We used the points of maximum occupation probability to determine the pinning sites of point-like hotspots directly.

Second, we used the averaged domain wall curvature illustrated in Extended Data Fig. 7d to localize the pinning sites of line-shaped hotspots. Namely, we extract extended areas with an averaged curvature larger than 1.5 µm$^{-1}$. Among those areas, we defined the point of the highest curvature as the location of the pinning sites in the line-shaped hotspots. To ensure a sufficient spatial separation, only pinning sites that are at least 8 pixels apart were considered.

### Comparison of CCI to existing time-resolved coherent imaging methods

In Extended Data Table 1 we compare CCI with existing time-resolved coherent imaging methods with magnetic contrast. The figures are based on the following considerations: Line 1: Results for conventional coherent imaging in this work. The diffraction-limited resolution of 18 nm is achieved when averaging over ~250 frames (see Methods section 'Real-space image reconstruction'), which corresponds to ~100 s. Of course, to achieve the best resolution, the system has to be static over the integration time. Line 2: Spatial and temporal resolution achieved in this work by employing CCI. To achieve the reported diffraction-limited spatial resolution, the same number of frames as in conventional imaging had to be averaged, but due to CCI without loss of temporal resolution. Statistical errors in the assignment of a state to a frame as well as uncertainties from grouping several internal modes into a state are not included in the resolutions specified and have to be considered additionally. Line 3: Anticipated potential of conventional coherent X-ray imaging with sub-10 nm resolution (a regime that is difficult to reach with noncoherent X-ray imaging techniques). The integration time drastically increases as the target spatial resolution increases because scattering signals typically decay strongly at high $q$. Line 4: CCI at synchrotron radiation sources. The frame rate reported here corresponds to the detector frame rate of the MOENCH detector[54], a particularly fast detector in the soft X-ray regime. Line 5: CCI at high-repetition-rate X-ray free-electron lasers. The perspective given here is based on the specifications of the European XFEL (EuXFEL)[55]. Time resolution is limited by the maximum bunch frequency (currently 1/(220 ns) at EuXFEL) and maximum duration is limited by the length of a bunch train (currently 600 µs at EuXFEL). Line 6: CCI in a pump–probe scheme at an XFEL, where the time resolution is limited by the pulse duration and the duration by the maximum pump–probe delay available. Line 7: Based on ref. [43]. Line 8: Based on refs. [56,57]. Line 9: Based on a combination of ref. [58], where nondestructive single-shot holographic magnetic imaging was demonstrated, and ref. [59], where it was shown that an image of the same specimen at two different times can be encoded in such a single-shot hologram.

## Data availability
The data reported in this study are available with identifiers at https://doi.org/10.5281/zenodo.7180988.

## Code availability
The code reported in this study are available with identifiers at https://doi.org/10.5281/zenodo.7180988.

**Acknowledgements** We thank C. Ropers, R. Comin, A. Menzel and D. Giannakis for discussions and advice. Work at MIT was supported by the DARPA TEE programme. Work at MBI was supported by the Leibniz Collaborative Excellence programme. Work at HZB was supported by the Helmholtz Young Investigator Group Program. This research used resources (beamline 23-ID-1, CSX) of the National Synchrotron Light Source II, a US Department of Energy (DOE) Office of Science User Facility operated for the DOE Office of Science by Brookhaven National Laboratory under contract no. DE-SC0012704.

**Author contributions** F.B., W.H., C.M., S.B.W., G.S.D.B. and B.P. conceived the project. F.B., I.L., M.S. and C.M.G. prepared the samples. C.K., F.B., W.H., I.L., J.M.B. and M.H. performed the experiments with support from C.M. and S.B.W.; C.K. analysed the data with supervision from B.P. and S.E.; R.B. performed the CDI reconstructions with support from S.Z., C.K., F.B., W.H., C.M. and B.P. interpreted the data with input from K.L., A.B. and S.B.W.; W.H., C.M., C.K., F.B. and B.P. developed the CCI technique. C.K., F.B., W.H., C.M. and B.P. wrote the paper with comments from all authors.

**Funding** Open access funding provided by Helmholtz-Zentrum Berlin für Materialien und Energie GmbH.

**Competing interests** The authors declare no competing interests.

**Additional information**
**Correspondence and requests for materials** should be addressed to Felix Büttner or Wen Hu.

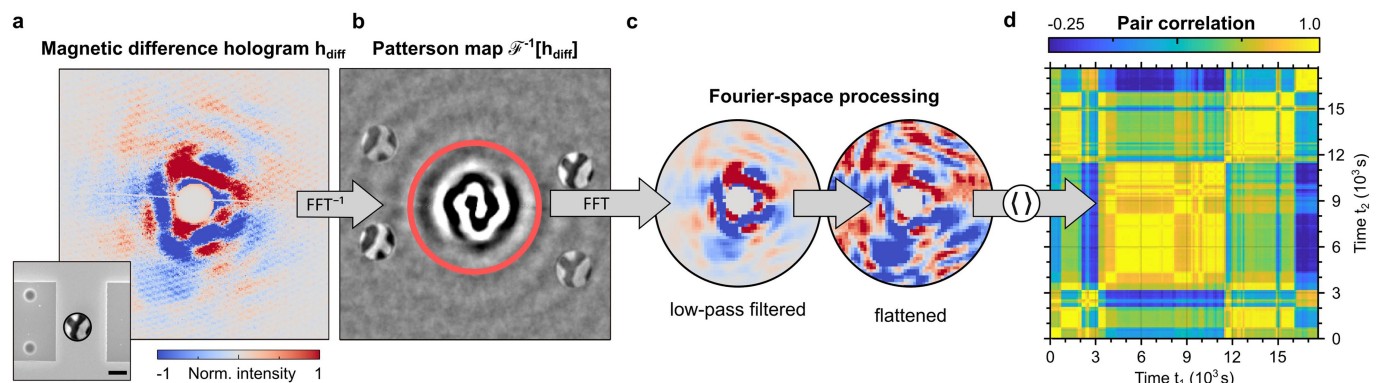

**a** Magnetic difference hologram $h_{diff}$

**b** Patterson map $\mathscr{F}^{-1}[h_{diff}]$

FFT$^{-1}$

FFT

**Fourier-space processing**

low-pass filtered

flattened

$\langle\rangle$

**c**

**d** −0.25   **Pair correlation**   1.0

−1   Norm. intensity   1

**Extended Data Fig. 1 | Algorithm to calculate the magnetic pair correlations map. a**, The calculation starts with the experimental data, that is, the magnetic difference hologram $h_{diff}(t)$ obtained at time $t$ from the difference of a circular polarization hologram and a topography hologram. Inset, scanning electron micrograph of the sample, superimposed by an image of the magnetic domain state. Scale bar, 500 nm. **b**, Patterson map obtained from an inverse Fourier transformation of $h_{diff}(t)$. The cross-correlation appears in the centre while the real-space magnetic-contrast image ('reconstruction') is offset from the centre by the object–reference distance. **c**, A low-pass filtered difference hologram is calculated from the Fourier transformation of the cross-correlation cropped from the Patterson map. Further normalization (flattening) is performed using the amplitudes of the topography scattering, i.e., the square-root of the reference-filtered topography hologram. **d**, Normalized pair correlation map calculated by the scalar product $\langle ., . \rangle$ of the flattened difference holograms (equation (1)). For illustration purposes, all panels are based on sequential averages of 100 hologram frames. See Methods for the mathematical steps.

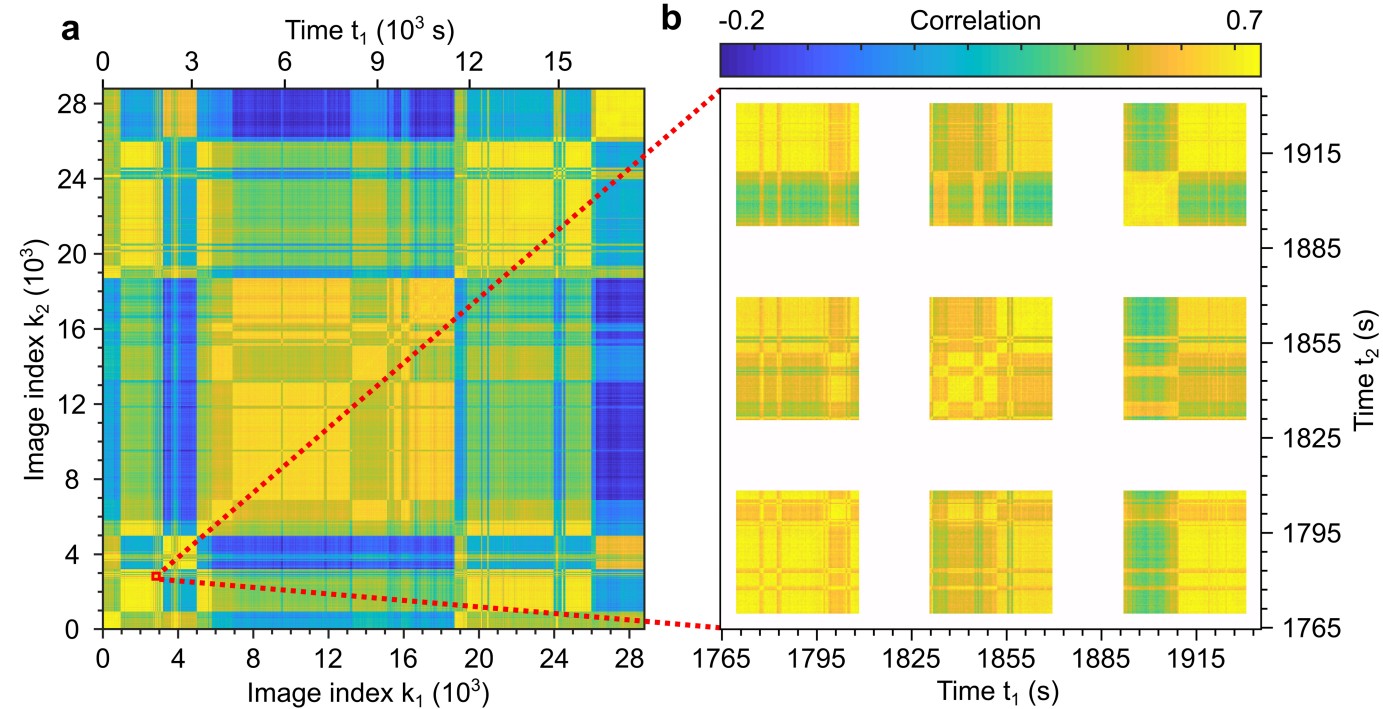

**Extended Data Fig. 2 | Complete pair correlation map. a**, Pair correlation map of all recorded frames (indexed sequentially). **b**, Magnified section of the correlation map, marked with a red square in **a**, now plotted as a function of the wall time (time stamp) when the respective frames were recorded. Each block represents a stack of 100 acquired frames. White areas indicate time intervals where no images were recorded, mostly due to helicity change events.

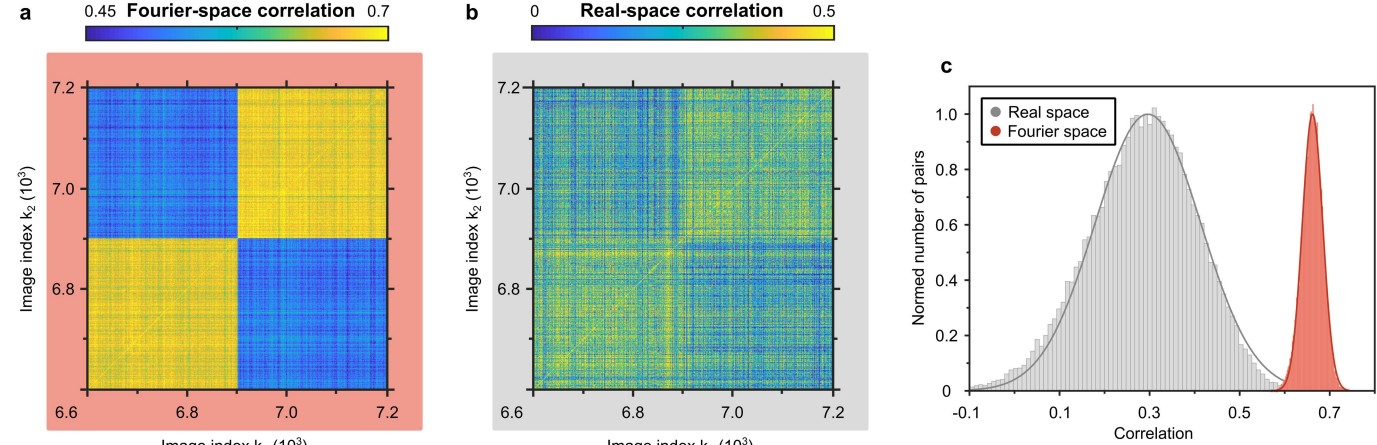

**Extended Data Fig. 3 | Signal-to-noise ratio of the real- and Fourier-space method of calculating correlation maps. a**, Fourier-space correlation map calculated with the algorithm presented in Extended Data Fig. 1, showing a transition between two states. **b**, The same correlation map calculated using single-frame real-space reconstructions (see Extended Data Fig. 1b). **c**, Histograms showing the distribution of observed pair correlation values.

The distributions were obtained from the same-state regions in each of the correlation maps (visible as yellow regions in **a**). To estimate the signal-to-noise ratio (SNR), both distributions were fitted with a Gaussian function. We define the SNR by the expectation value divided by the standard deviation of the fit. We find that the Fourier-space SNR is more than 10 times higher ($29.3 \pm 1.0$) than the real space SNR (SNR $2.5 \pm 0.2$).

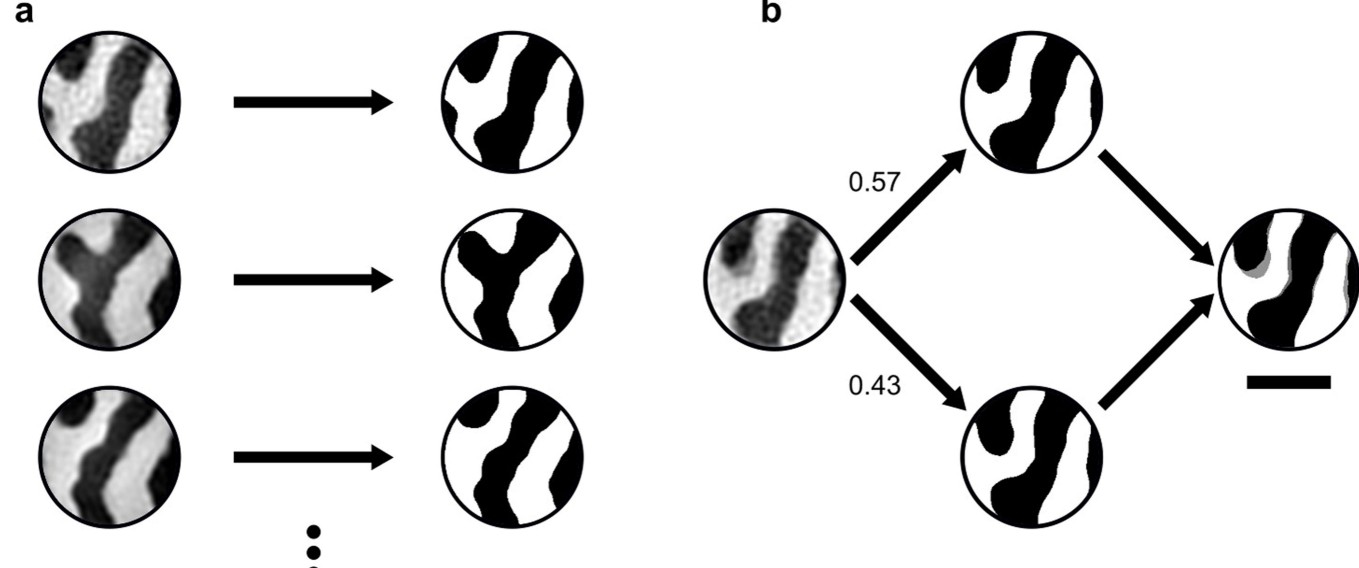

**Extended Data Fig. 4 | Discretization of the internal domain modes.**
**a**, Creation of binary internal domain modes. The up and down magnetization of all domain images is binarized using image segmentation based on threshold filtering of the CDI reconstructions. The representations of images with a low signal-to-noise ratio were adjusted manually. **b**, Illustration of the multi-level discretization of internal domain modes. The CDI reconstruction of a domain image is decomposed into a set of binarized internal modes. The binary images of the internal modes are superimposed to create a discretized representation of the domain image with multiple contrast level. A fitting process determines the weighting coefficients from the image of the domain mode. For further details about the procedure, see Methods. Scale bar, 500 nm.

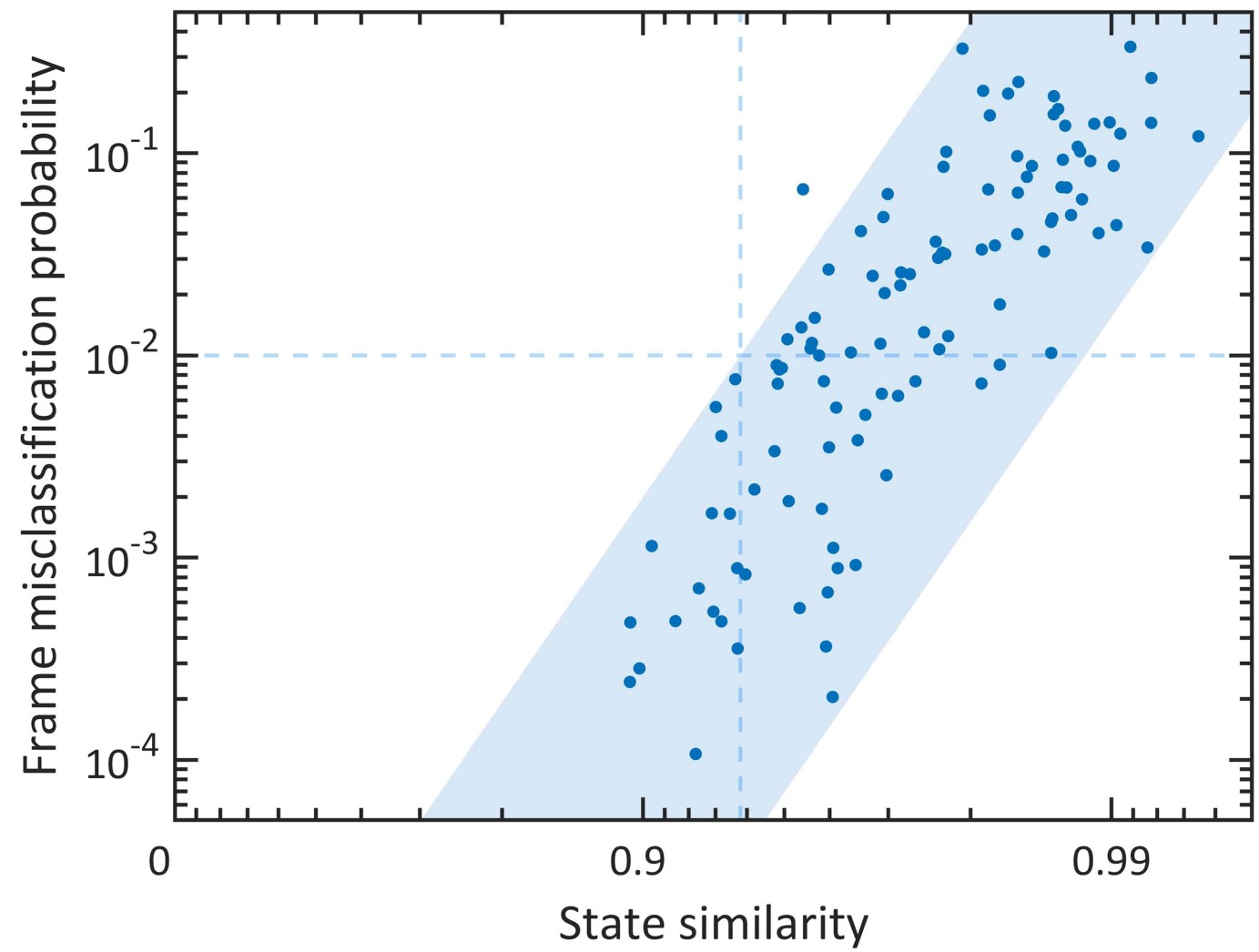

**Extended Data Fig. 5 | Sensitivity limit of the temporal reconstruction.** Frame misclassification probability of a two-state system as a function of the similarity of the underlying two domain states. A similarity value of one corresponds to indistinguishable states. Data points below 87% similarity show a misclassification probability of zero and were therefore excluded from the log-log plot. The light-blue band is a guide to the eye. Our sensitivity threshold is marked by the dashed blue lines. The horizontal line indicates the largest acceptable misclassification rate for temporal assignment (1%). The vertical line depicts the maximum similarity of two states before and after a temporally resolved transition (93.8%), which was chosen such that all data points left of this line are below the misclassification threshold. For further details, see Methods.

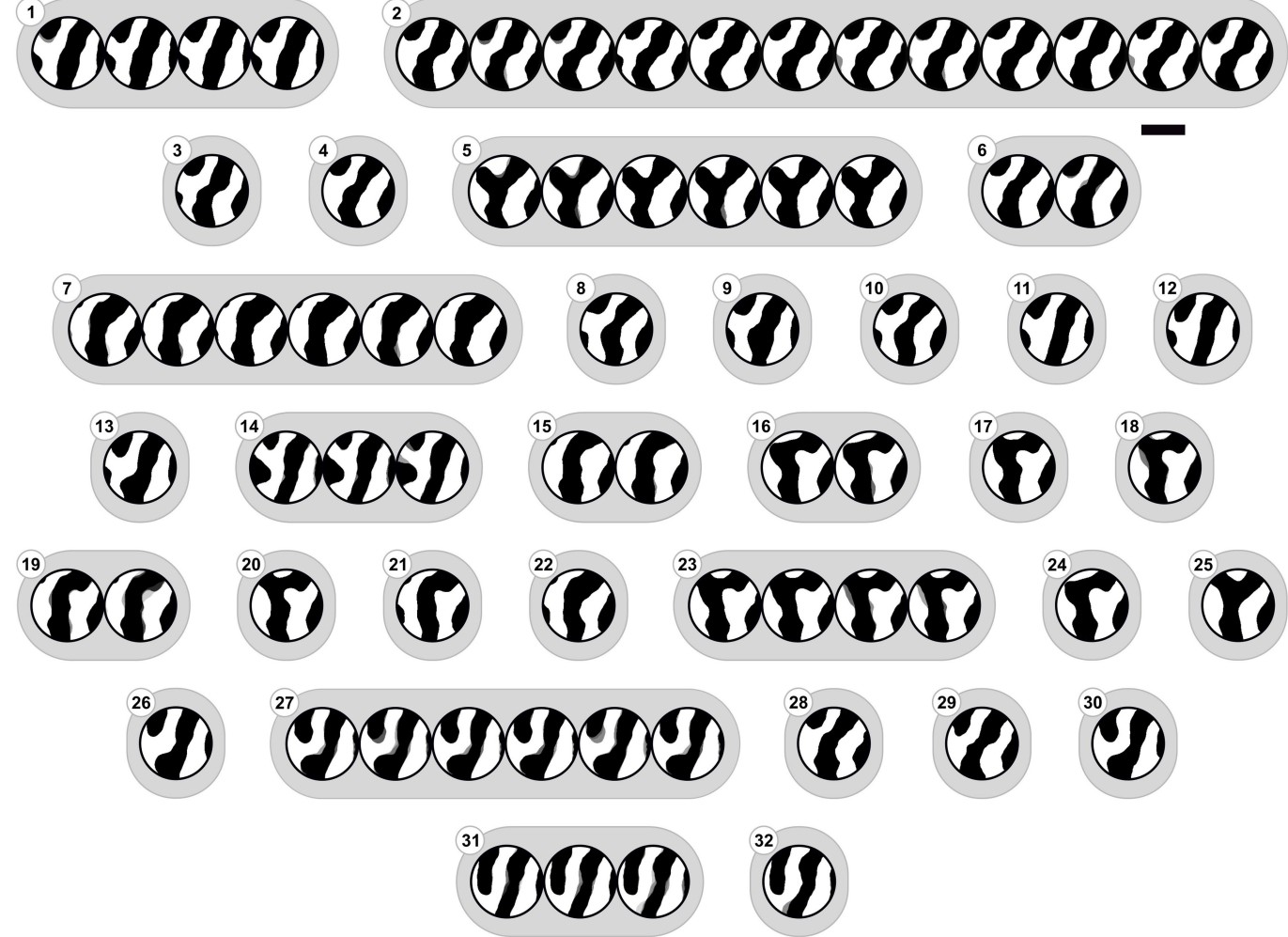

**Extended Data Fig. 6 | Overview of the reconstructed discretized internal modes.** We identified a total of 72 internal modes, ranging up to 99% in similarity (see Methods). The modes are subsets of the 32 domain states (see Fig. 4), as indicated by the grey background shading. Modes are defined by a high pair correlation of >93.8%, which means that their exact temporal sequence is inaccessible but statistical information, such as the real-space images and the number of contributing frames, can still be reconstructed reliably. Scale bar, 500 nm.

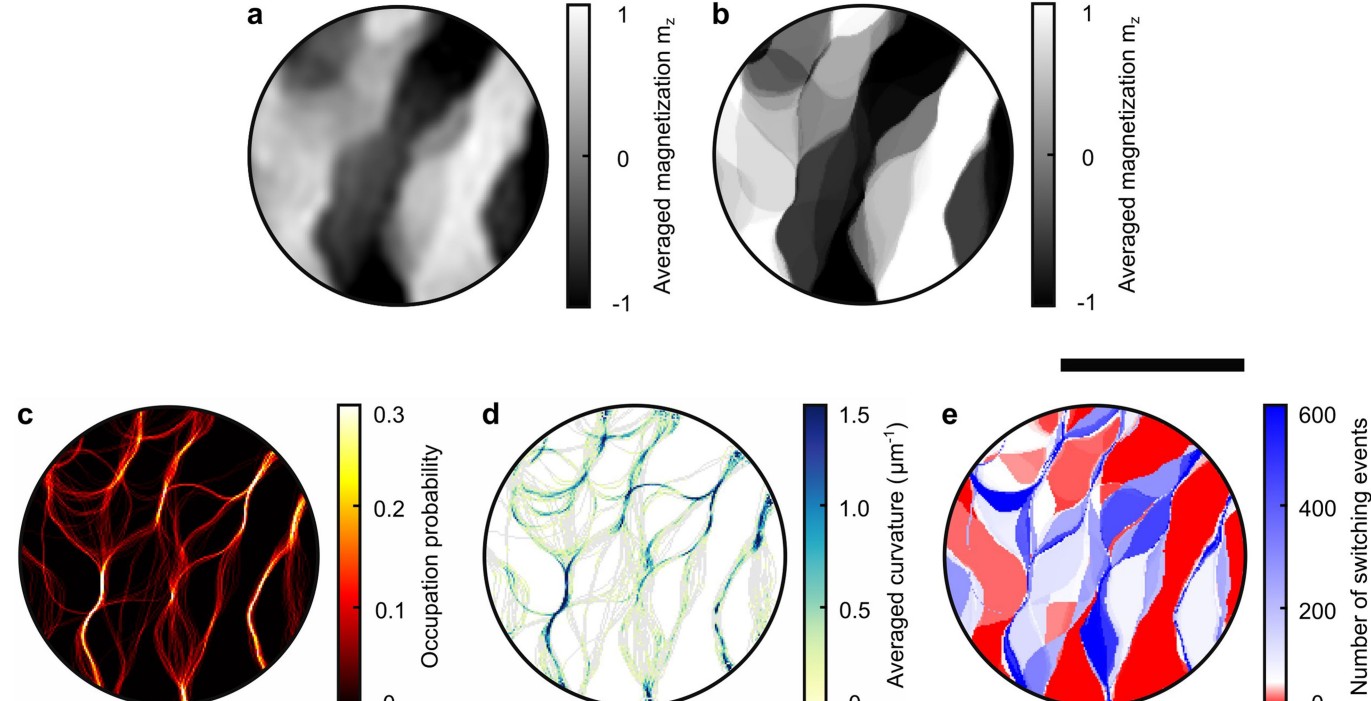

**Extended Data Fig. 7 | Statistical analysis of the real-space reconstructions. a**, Average of all domain configurations observed during the experiment weighted according to the number of frames for each configuration. **b**, Similar average of the discretized domain configurations. **c**, Superposition (sum) of the domain wall positions of all internal modes, normalized by the total number of modes. **d**, Average curvature of the domain walls. Only walls showing a considerable curvature are visualized in colour. Grey lines mark remaining domain walls. **e**, Number of magnetization switching events observed locally in the temporal evolution of the domain configuration. Scale bar, 500 nm.

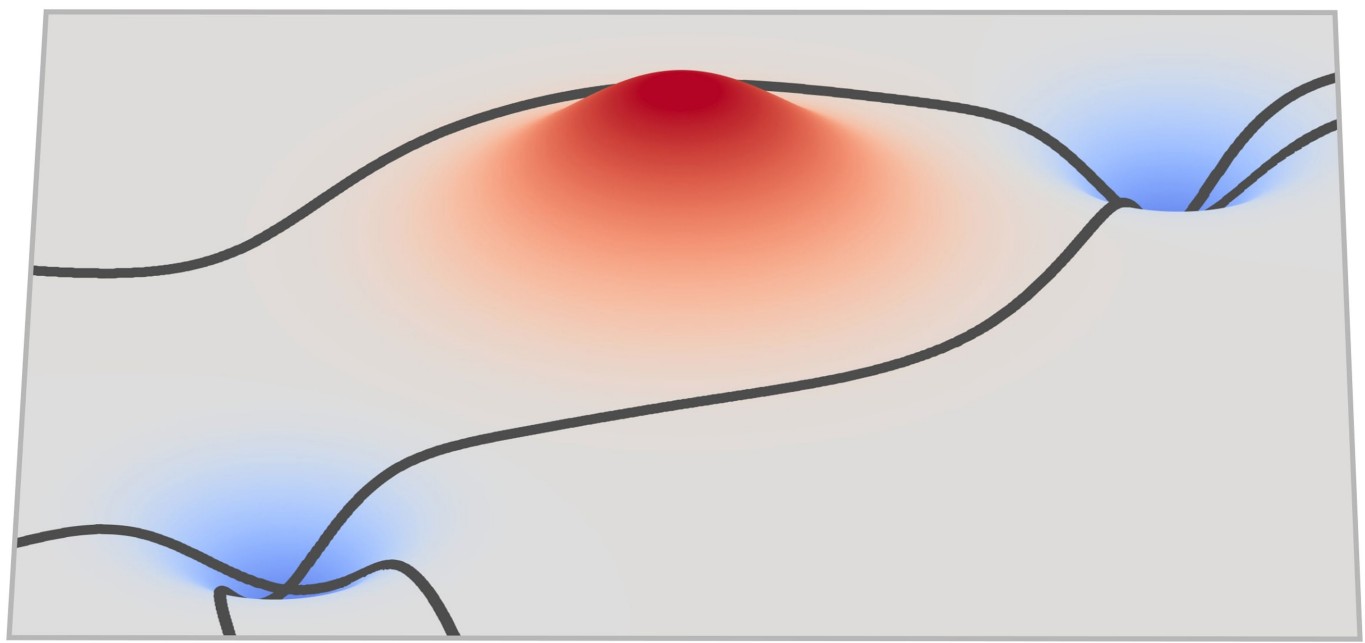

**Extended Data Fig. 8 | Illustration of the conceptual difference between attractive and repulsive pinning.** String-shaped magnetic domain walls (grey lines) located in an energy landscape with attractive and repulsive Gaussian-shaped pinning potentials. The attractive pinning sites (blue) induce depressions in the energy landscape trapping the domain walls at their minimum, whereas the repulsive pinning site (red) forms an area-like potential barrier repelling the domain walls.

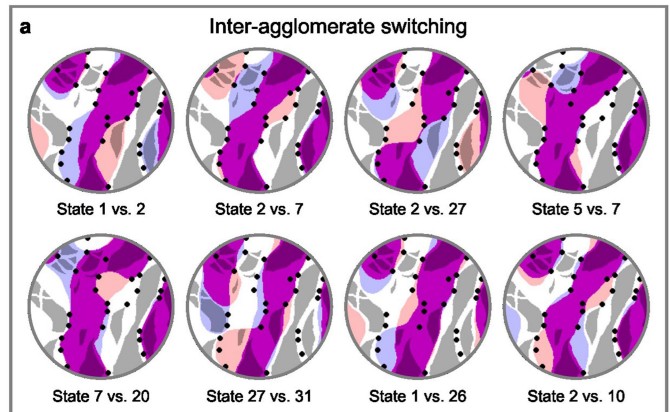

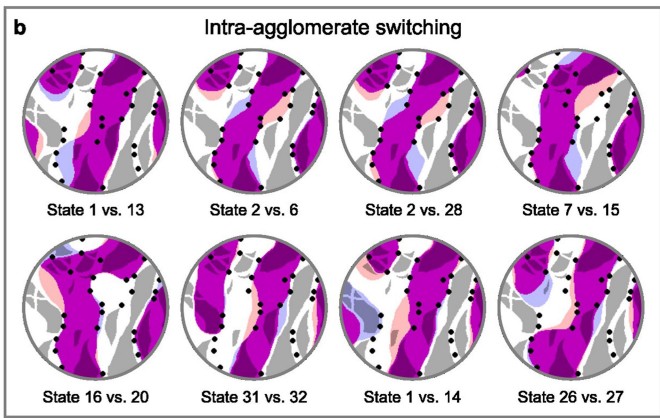

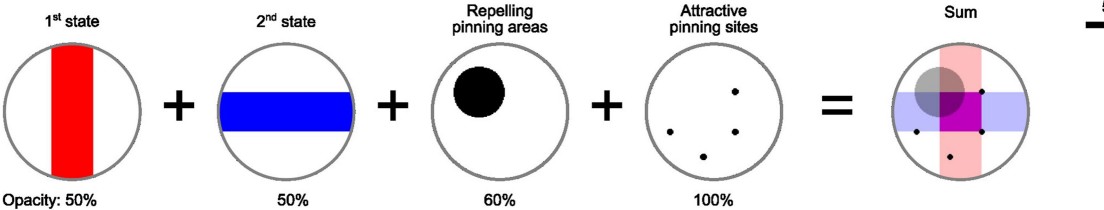

**Extended Data Fig. 9 | Domain-wall rearrangement during transition between two states. a,b,** Selected examples for inter-agglomerate (**a**) and intra-agglomerate (**b**) transitions. The colour code is explained below both panels. The summed contrast shows areas without magnetization changes as purple or white and areas with switched magnetization as blue or red. Areas containing repelling pinning sites are marked with semi-transparent grey shadows, whereas attractive pinning sites are marked with black dots. Scale bar, 500 nm.

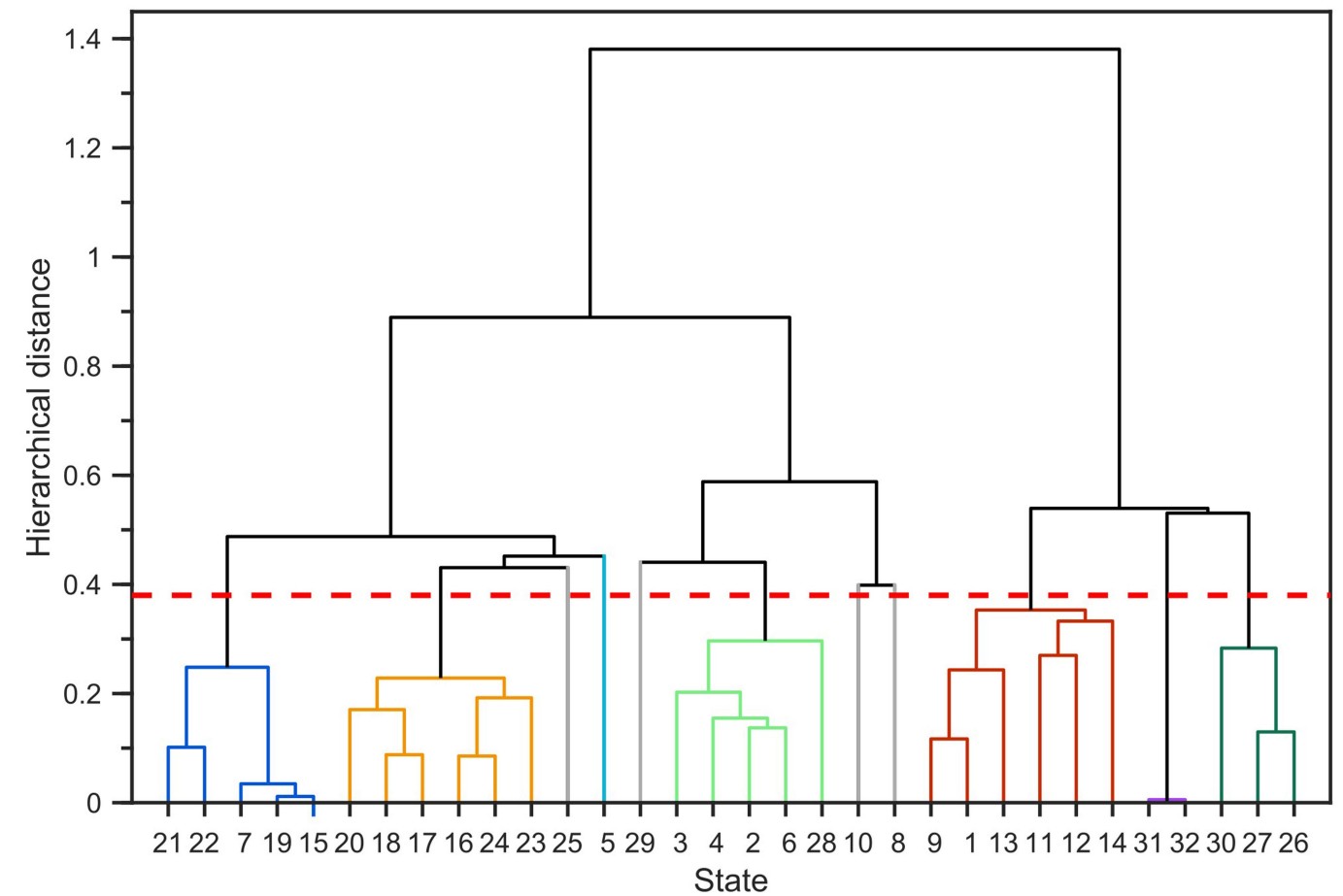

**Extended Data Fig. 10 | Dendrogram of state agglomeration.** An UPGMA hierarchical clustering algorithm groups the 32 domain states shown in Fig. 4 into 11 agglomerates based on the state's similarity distances and the frequency of transitions between the states. The separation threshold was manually set at a generally large step in the hierarchical distance (0.38) as indicated by the dashed red line. The colour coding separates the different agglomerates and corresponds to Fig. 4. Single, short-lived states which are not colour-coded in Fig. 4 are shown in grey here.

**Extended Data Table 1 | Comparison of CCI to existing time-resolved coherent X-ray imaging methods**

| Experimental Method | Spatial resolution | Time resolution | Maximum duration | Type of dynamics |
|---|---|---|---|---|
| **This Work** | | | | |
| Conventional coherent imaging | 18 nm | 100 s (250 frames) | unlimited | *Stochastic* |
| CCI | 18 nm | 387 ms | unlimited | *Stochastic* |
| **Perspectives** | | | | |
| Conventional coherent imaging at current SR sources | <10 nm | > 1 h | unlimited | *Stochastic* |
| CCI at current SR sources | <10 nm | 1 ms | unlimited | *Stochastic* |
| CCI at current high-rep-rate XFELs (here: EuXFEL) | <10 nm | 220 ns | 600 μs | *Stochastic* |
| Pump-probe CCI | <10 nm | <10 fs | unlimited | Triggerable *stochastic* |
| **Literature** | | | | |
| Pump-probe coherent imaging at SR sources | 40 nm | 50 ps | ≈1 μs | Triggerable *repeatable* |
| Pump-probe coherent imaging at XFELs and HHG sources | 40 nm | <10 fs | unlimited | Triggerable *repeatable* |
| Single-shot X-ray coherent imaging at XFELs | 80 nm | 80 fs | two frames | *Stochastic* |

The first section provides the direct comparison of CCI and conventional coherent imaging (phase retrieval of blindly averaged frames) based on the data of the present experiment. The second section illustrates the anticipated potential of CCI based on specifications of sources[55] and detectors[54] already available today. The last section highlights achievements of previous time-resolved coherent imaging experiments, which have focused on repeatable[43,56,57] dynamics or on single-shot measurements[58,59]. For the sake of comparability, the table focuses on magnetic imaging. For further details and references, see Methods.