## [Peer review file · Nature]

Manuscript Title: Coherent Correlation Imaging: Resolving fluctuating states of matter

Reviewer Comments & Author Rebuttals

Reviewer Reports on the Initial Version:

Referees' comments:

Referee #1 (Remarks to the Author):

Klose and coworkers have developed a methodology for a direct observation of nanometer-scale fluctuation in a temporal resolution of sub-second. In the introduction part, they mention the fundamental trade-off between a high spatial resolution and a high temporal resolution. Then they briefly explain the key concept of 'coherent correlation imaging'. The data acquisition seems to be essentially identical to the x-ray photon-correlation spectroscopy. The series of speckle images are classified into several patterns by a method similar to the gene identification. The conversion from the reciprocal to the real space is based on the holography-type phase retrieval. This method is applied to the spatiotemporal observation of magnetic domains in a thin film of a perpendicular magnetic ferromagnet. The field of view is circular with a diameter of 720 nm. Two reference holes of 40-nm and 60-nm diameters are used for the phase retrieval. The coherent circularly polarised soft x-ray beam of a photon energy of Co L3 edge is irradiated to the sample. The polarisation is reversed every 100 hologram frames. The FFT image of the scattered intensity distribution comprises the cross-correlation image at centre and four holographic images. The acquired 28800 cross-correlation images are classified into 36 'states', and the real-space image of each 'state' is reconstructed by much better statistics than in the case of single-frame holography. A unique stochastic temporal evolution is then easily revealed. Moreover, the 36 'states' are clustered into 11 groups by using the UPGMA.

I completely agree with the authors that the observation of random dynamics of an aperiodic distribution is a next-generation key technology. I think that the present methodology should be highly esteemed. The proposed 'coherent correlation imaging' is composed of several cutting-edge techniques and the presentation is logical enough to be approved.

I recommend the publication of this work in Nature after some minor revisions. The authors may consider the following points.

1. As the authors state in the introductory part, there is generally a trade-off between the spatial temporal resolutions. The proposed method is a clever way to secure the efficient averaging time long enough for a high spatial resolution with a shorter acquisition time (temporal resolution). Nonetheless, there should still exist a limit of the temporal resolution. The authors should discuss how the temporal resolution is improved by the present method from the 'conventional' single-shot holography and the limit of the temporal resolution exemplifying the magnetic domain observation of the present target.

2. The grouping of cross-correlation images into a small number of 'states' is essential for the

present method. I suppose that the images should be classified in a more detailed manner to improve the spatial resolution. In this study, probably, a threshold of pair correlation is introduced for the grouping. How is the threshold value optimised? In other words, the authors may discuss the relation between the acquisition time for each frame and the group number. This would give the temporal resolution of the proposed method.

3. How is the size of field of view designed? The optimum size is the same as in the case of single-shot coherent imaging?

4. The obtained dynamics would indicate some correlation between the jumps of neighbouring domain walls. Are the simultaneous jumps of different domain walls real or an artefact arising from the clustering?

5. I am afraid that in 'Robust classification of frames' the Figure number is not correct.

6. The colours for states 10 and 14 in the dendrogram in Fig. E10 do not correspond to that in Fig. 4.

Referee #2 (Remarks to the Author):

The paper presents a new imaging approach, coherent correlation imaging (CCI), which looks especially promising for studying fluctuations and stochastic events in the system. The approach may allow one to improve the temporal resolution of the sample reconstructions, without sacrificing spatial resolution. This is achieved by using a dedicated data classification scheme, based on finding, amplifying and quantifying correlations between the measured noisy images. Such classification allows to attribute each measured scattering image to a specific discrete state of the system. By applying assisted averaging of the classified data one can obtain high-resolution real-space images of distinct states, which might be otherwise hidden in the usual (non-assisted) temporal average.

The authors demonstrated their approach in the study of domain-wall fluctuations in a magnetic system with highly degenerate ground state. By applying CCI they discovered that the spatiotemporal evolution of the magnetic system under study can be indeed described by a final number of different magnetic states of the system. The detailed temporal analysis of transitions between different states revealed complex non-evolving character of dynamics, which appears to be incompatible with previously reported thermally activated processes. Based on the information accessible from CCI the authors devise a physical picture of the complex dynamics in the system, described in terms of interplay between attractive and repulsive pinning. Therefore, the outlined approach may provide access to new information about the stochastic dynamics in the system, typically hidden in the temporal averages applied in more conventional approaches.

The presented results are novel and of general interest for broad community. It would be certainly exciting to look also at other systems. The manuscript deserves publication in Nature after the following comments/criticism are considered by the authors:

(1) The choice of specific parameters in the data classification part of the approach looks quite

arbitrary, (e.g., misclassification threshold, maximum similarity allowed between images in different clusters, etc.). Some of these values are not well justified. Their variation may change some of the results, particularly those shown in Fig. 4.

(2) Based on the above, the authors should be careful about some conclusions. For instance, fast oscillatory hopping between states shown in the temporal evolution in Fig. 4b may be affected by a specific choice of the above mentioned parameters.

(3) Challenge in application to other systems might be the scalability of the approach, i.e., number of frames required to record more than the number of states considered in the present study.

There are some misprints in the manuscript:

(1) Page 20, lines 550-551, should be "(panel d)", instead of "(panel h)"

(2) Pages 20-21, in Section "Robust classification schemes" one has to replace "Fig. 2" everywhere by "Fig. 3".

Referee #3 (Remarks to the Author):

Rejected on grounds of specialist interest, lack of novelty, insufficient conceptual advance

The authors present as novel a technique they branded CCI Coherent Correlation Imaging, to image time dependent magnetic fluctuations.

Since the high brightness new sources became popular, Coherent Diffraction Imaging (CDI) has been at the forefront of the experiments using the coherence of X-rays for imaging. Meanwhile X-ray Photon Correlated Spectroscopy (XPCS) has been used to observe the dynamics and fluctuations in materials. Differences in the experimental set-ups, especially the detector specifications and the required coherent photon flux conditions, didn't allow the simultaneous, dual, spatial and temporal analysis of the scattered patterns recorded on those experiments. Meaning that the analysis of the XPCS data didn't allow for the reconstruction of images when performing XPCS, nor extract time dependences of the CDI images, most of it because of the need for long time exposures for good signal to noise ratios.

Scientists have been creative in how to overcome these limitations, until approximately 8 years ago, a new generation of faster area detectors provided that possibility. CSX beamline has one of these detectors. Increasing the detectors recording speed while having enough photon flux allows faster collection of data frames and make the incursion in the faster time regimes. This was mainly explored at the new emerging coherent sources, such as FELs.

Work has been done in the past that shows magnetic domains configurations, showing pinning points along the different magnetic states over a magnetization curve. Recently a lot of efforts has been put in showing the dynamics of skyrmions with some of the experiments been successful.

The novelty presented in this work resides mainly in the imaging reconstruction algorithm. How Fourier images are treated to finally obtain the images of magnetic domains, correlated with temporal information. The principle of the algorithm used, has been tested before with other type of samples, what is OK and provides the right background.

The recommendation is not to publish in Nature, because the interest is very specialized, not enough novelty. The referee does not denies that this could be a new tool in general to study via imaging fluctuations in materials. However the lack of quantitative comparison, in terms of spatial-temporal resolution with other already established methods or experimental techniques, makes it difficult to foresee the improvement that CCI would bring to the field.

Furthermore, it is missing for an article aspiring to present a novel technique, a quantitative evaluation of the limits of the technique in terms of coherent photon flux (max / min) requirements for the application of CCI, the proportion of partially coherent photons, and under which beam and experimental conditions it could be improved and make more general than for the case presented.

No resubmission is recommended.

Author Rebuttals to Initial Comments:

Reply to the Referees

We would like to thank all referees for their detailed and very helpful feedback on our manuscript. In particular, we thank reviewers #1 and #2 for their positive and, at the same time, critical reviews. Their feedback has inspired us to thoroughly take into account the influence of the parameters of our algorithm on the temporal resolution, and to better explain the general steps and limitations leading to the emergence of temporal resolution in the main text. This feedback, and all other comments, have been fully considered in the revision of the manuscript.

The most important changes are:

- A change of the criterion to decide in the iterative clustering (Fig. 3c) whether the dendrogram can be separated into two states or not. Where before we applied a manual, visual criterion, we now replaced this with a numeric one based on the “inconsistency parameter ξ ”, as defined in the Methods and in Fig. 3c.
- Based on the new, fully numerical algorithm, we tested the robustness of the results obtained for varying thresholds for ξ . Importantly, within a broad range (from 1.7 to 2.2), the threshold for ξ only changes the number of internal modes and states but none of the physics that we extract from the data (i.e., Fig. 4 remains essentially invariant under these changes). In a way, ξ can be seen as a “binning parameter”, which defines how finely or coarsely the clustering and grouping is done, but not affecting the spatio-temporal resolution.
- In this context, the number of states and modes has changed (before: 36 and 86, now: 32 and 72) without affecting Fig. 4 in a relevant way.
- We have reworked our description of the algorithm in the main text to explain the emergence of spatial and temporal resolution more clearly. This includes a modification of Fig. 3 to include the step of “grouping”, which needs to be performed to reconstruct spatio-temporally resolved “states” from just-spatially-resolved “internal domain modes”.
- The Methods section and all affected Figures have been updated accordingly, with all changes highlighted in the tracked-changes document.

In the following we reply to the individual comments of the referees point-by-point. We reprint the referees' feedback in normal font and our reply in bold.

Reviewer #1 (Remarks to the Author):

Klose and coworkers have developed a methodology for a direct observation of nanometer-scale fluctuation in a temporal resolution of sub-second. In the introduction part, they mention the fundamental trade-off between a high spatial resolution and a high temporal resolution. Then they briefly explain the key concept of 'coherent correlation imaging'. The data acquisition seems to be essentially identical to the x-ray photon-correlation spectroscopy. The series of speckle images are classified into several patterns by a method similar to the gene identification. The conversion from the reciprocal to the real space is based on the holography-type phase retrieval. This method is applied to the spatiotemporal observation of magnetic domains in a thin film of a perpendicular magnetic ferromagnet. The field of view is circular with a diameter of 720 nm. Two reference holes of 40-nm and 60-nm diameters are used for the phase retrieval. The coherent circularly polarised soft

x-ray beam of a photon energy of Co L3 edge is irradiated to the sample. The polarisation is reversed every 100 hologram frames. The FFT image of the scattered intensity distribution comprises the cross-correlation image at centre and four holographic images. The acquired 28800 cross-correlation images are classified into 36 'states', and the real-space image of each 'state' is reconstructed by much better statistics than in the case of single-frame holography. A unique stochastic temporal evolution is then easily revealed. Moreover, the 36 'states' are clustered into 11 groups by using the UPGMA.

I completely agree with the authors that the observation of random dynamics of an aperiodic distribution is a next-generation key technology. I think that the present methodology should be highly esteemed. The proposed 'coherent correlation imaging' is composed of several cutting-edge techniques and the presentation is logical enough to be approved.

I recommend the publication of this work in Nature after some minor revisions. The authors may consider the following points.

1. As the authors state in the introductory part, there is generally a trade-off between the spatial temporal resolutions. The proposed method is a clever way to secure the efficient averaging time long enough for a high spatial resolution with a shorter acquisition time (temporal resolution). Nonetheless, there should still exist a limit of the temporal resolution. The authors should discuss how the temporal resolution is improved by the present method from the 'conventional' single-shot holography and the limit of the temporal resolution exemplifying the magnetic domain observation of the present target.

So far, only few studies have demonstrated imaging of magnetic structures by single-shot holography, because the extremely bright x-ray pulses needed to encode a full magnetic image in a single shot perturb, degrade, or destroy almost any sample. Even in the small fluence window where non-destructive single-shot magnetic imaging is possible [1,2] (for some selected extremely high contrast materials), the resulting image quality from a single shot is rarely sufficient to draw scientific conclusions. From our results it can be projected that if CCI is applied to single-shot data, multi-frame averaging becomes possible without any compromise to the single-shot temporal resolution – with ~100 times lower incident flux, i.e., robustly in the non-perturbative regime for a wealth of materials. Hence, while the temporal resolution of single-shot imaging is not improved by CCI (it is already as good as it can be!), CCI instead removes the so-far prohibitive drawback of single-shot imaging, namely the extremely poor spatial resolution, making FEL-based imaging broadly applicable to condensed matter research.

[1] Wang, T. et al. Femtosecond Single-Shot Imaging of Nanoscale Ferromagnetic Order in Co/Pd Multilayers Using Resonant X-Ray Holography. *Physical Review Letters* 108, 267403 (2012).

[2] von Korff Schmising, C. et al. Imaging Ultrafast Demagnetization Dynamics after a Spatially Localized Optical Excitation. *Physical Review Letters* 112, 217203 (2014).

Considering that our results were not acquired at a free-electron laser source, we believe that the discussion on the topic of single-shot imaging in the manuscript should be brief. We have therefore condensed the discussion above to the following sentence at the end of the discussions, which now emphasizes the gain in spatial resolution for single-shot imaging:

“In conjunction with the next generation of free-electron x-ray lasers running at MHz repetition rates⁹, real-time imaging with sub-microsecond temporal resolution as well as non-

destructive pump-probe imaging with femtosecond resolution and wavelength-limited spatial resolution will be feasible.”

Moreover, we have also added to the results section (page 5, discussion of Fig. 2) a description of parameters that affect the temporal resolution of CCI:

“On the contrary, temporal resolution emerges from the precision by which each snapshot, and hence timestamp, is assigned to the correct state. The temporal resolution of CCI is hence independent of the spatial resolution, i.e., the captured q -range, but instead depends on the similarity between states and on the speckle contrast.”

2. The grouping of cross-correlation images into a small number of 'states' is essential for the present method. I suppose that the images should be classified in a more detailed manner to improve the spatial resolution. In this study, probably, a threshold of pair correlation is introduced for the grouping. How is the threshold value optimised? In other words, the authors may discuss the relation between the acquisition time for each frame and the group number. This would give the temporal resolution of the proposed method.

We agree with the referee that the assignment of frames to the small number of states is essential for the spatio-temporal reconstruction. We have taken this comment and the comment of referee #2 as a reason to revisit the classification part by our clustering techniques and evaluate the choice of specific clustering parameters. In the following we feature the essential aspects of the three-phase classification procedure.

First, we apply an iterative clustering algorithm to identify all domain configurations occurring in the experiment based on the pair-correlation between all frames. This is now illustrated in Fig. 3a-d. Here, the key is to iteratively split clusters that show a natural division, i.e., evidence of a superposition of multiple domain states. The result is a set of inseparable clusters, which we call “internal domain modes”, each representing a distinctly different domain configuration.

To quantify the previously manual evaluation of whether a cluster “shows a natural division”, we made the following changes to our algorithm:

- We now differentiate natural divisions in the clusters by thresholding of a single quantity – the inconsistency coefficient ξ – which is a common measure of dissimilarity in hierarchical clustering.
- For optimization, we analyzed the impact of different thresholds on the frame misclassification probability and chose the threshold $\xi = 1.85$ as it produces the lowest error.

In the second phase we estimate the misclassification error and set a threshold. This part is discussed in detail in the reply to referee #2.

And in the third phase, we group our set of internal domain modes into the small number of states, such that the misclassification error between frames and states is below the target threshold. Because of this low error, the assignment between frames and states becomes invertible, i.e., one can make the relation between the group number and the acquisition time, as pointed out by the referee.

Our new manuscript now discusses the choice of parameters of this process, i.e., what the parameters are, why their values were chosen, and how a different choice would influence the temporal resolution.

Specifically, we modified Fig. 3 and the corresponding discussion in the main text to include these important, temporal-resolution-defining steps now also in the main body of the manuscript. Moreover, we have modified the Methods, specifically the sections “*Robust classification of frames*”, “*Reconstruction of internal modes*”, “*Estimation of the temporal discrimination threshold and reconstruction of the 36 states*” and “*Determination of agglomerates in the transition network*” to reflect the new technical implementation of the algorithm. We also updated Figs. 1 and 4, Extended Figs. E4 and E5 as well as Extended Figs. E6, E9 and E10 according to the revisited frame classification.

The resulting changes are:

- The number of identified domain modes decreased from initially 85 to 72 modes.
- The highest mode similarity allowing a frame misclassification probability below 1% decreased from initially 96% to 94% similarity.
- The number of states reduced from initially 36 to 32 due to the lower similarity threshold. However, states are still assigned to 99.5% of the 28800 frames.

We emphasize that these changes have no impact on our conclusions.

In summary, we can show that with the chosen signal strength in our experiment, we can find spatiotemporally resolved states with high accuracy. Specifically, we can reconstruct the temporal evolution between domain configurations with 99% accuracy, even if they show an extremely high similarity up to ~94% which is key for the temporal reconstruction of the dynamics. Here, the temporal resolution is given by the acquisition time of a single frame which is ~0.4 s. Smaller acquisition times have not been investigated.

3. How is the size of field of view designed? The optimum size is the same as in the case of single-shot coherent imaging?

The size of the field of view reflects a compromise between sensitivity for small changes and efficient use of the limited dynamic range of the camera (as favored by smaller object holes) and the need to gain contextual information for which larger fields of view are needed. The specific choice of the sample geometry is based on our experience with time-resolved x-ray holography of very similar magnetic materials. The corresponding explanation has been added to the end of the “Sample fabrication” section of the Methods.

4. The obtained dynamics would indicate some correlation between the jumps of neighbouring domain walls. Are the simultaneous jumps of different domain walls real or an artefact arising from the clustering?

We are convinced that the correlation between jumps of different domain walls is real and not an artefact arising from the clustering.

First, from the methodological point of view, there is no explicit or implicit entering of the “synchronization” of domain wall displacements in any step of the algorithm.

And second, the simultaneous jumps are easily explained by the physics of stripe domains: the competition of domain wall energy cost and stray field energy gain defines an optimum width for a domain. Hence, if one domain wall moves, the others tend to follow to (approximately) retain the domain width. We have added a clarifying comment to the end of the section “Estimation of the temporal discrimination threshold and reconstruction of the 32 states” in the Methods:

“Note in this context that the correlation between domain wall hopping in different parts of the field of view is not an artefact of the clustering algorithm but rather emerging from the intrinsic physics of the material, namely from the desire of the material to retain a constant domain width as set by the competition of domain wall energies and long-range stray field energies.”

5. & 6. I am afraid that in 'Robust classification of frames' the Figure number is not correct. The colours for states 10 and 14 in the dendrogram in Fig. E10 do not correspond to that in Fig. 4.

We thank the referee for pointing out these mistakes, which we corrected.

Reviewer #2 (Remarks to the Author):

The paper presents a new imaging approach, coherent correlation imaging (CCI), which looks especially promising for studying fluctuations and stochastic events in the system. The approach may allow one to improve the temporal resolution of the sample reconstructions, without sacrificing spatial resolution. This is achieved by using a dedicated data classification scheme, based on finding, amplifying and quantifying correlations between the measured noisy images. Such classification allows to attribute each measured scattering image to a specific discrete state of the system. By applying assisted averaging of the classified data one can obtain high-resolution real-space images of distinct states, which might be otherwise hidden in the usual (non-assisted) temporal average.

The authors demonstrated their approach in the study of domain-wall fluctuations in a magnetic system with highly degenerate ground state. By applying CCI they discovered that the spatiotemporal evolution of the magnetic system under study can be indeed described by a final number of different magnetic states of the system. The detailed temporal analysis of transitions between different states revealed complex non-evolving character of dynamics, which appears to be incompatible with previously reported thermally activated processes. Based on the information accessible from CCI the authors devise a physical picture of the complex dynamics in the system, described in terms of interplay between attractive and repulsive pinning. Therefore, the outlined approach may provide access to new information about the stochastic dynamics in the system, typically hidden in the temporal averages applied in more conventional approaches.

The presented results are novel and of general interest for broad community. It would be certainly exciting to look also at other systems. The manuscript deserves publication in Nature after the following comments/criticism are considered by the authors:

1. The choice of specific parameters in the data classification part of the approach looks quite arbitrary, (e.g., misclassification threshold, maximum similarity allowed between images in different clusters, etc.).

Some of these values are not well justified. Their variation may change some of the results, particularly those shown in Fig. 4.

We agree with the referee that specific parameters in the classification are crucial for the results of our analysis and that their choice needs to be clarified. As already stated in the reply to referee #1, we have taken the comment as a reason to revisit the classification part by our clustering techniques and evaluate the choice of specific clustering parameter.

An overview of the revision was already given in the answer to referee #1. Here, we will go into more detail and focus specifically on the choice of parameters in the classification. We recall that our classification is a three-phase procedure.

In the first phase, we apply an iterative clustering algorithm to successively separate our data set of frames until, finally, a set of inseparable pure-state cluster is derived. In our new version, the inconsistency coefficient is the crucial parameter which differentiates the desired pure-state clusters from mixed-states clusters, i.e., clusters that show evidence of a superposition of states and need to be reiterated.

We modified Fig. 3 to illustrate the quantities that enter into the newly introduced inconsistency parameter (the step height and the standard deviation) and made the following changes to the methods section “Robust classification of frames” to clarify its use in the iterative hierarchical clustering algorithm.

“Our third iteration step is to decide if our two identified sub-clusters represent pure or mixed-states, i.e., if they comprise frames from a single state or multiple states. To this end, we quantify the distinctness of the top-most link in the dendrogram by the inconsistency coefficient ξ , where ξ is given as the ratio of the average absolute deviation of the link step size to the standard deviation of all lower-level links included in the calculation. On this basis, we identify all clusters that undercut an inconsistency threshold as pure-state clusters while, in complementary, mixed states are present in clusters exceeding this threshold.”

The inconsistency threshold is linked to the spatial sensitivity of the clustering because, typically, more dissimilar domain configurations are indicated by larger hierarchical distances of nodes in the dendrogram. To justify a specific choice of inconsistency value, we analyzed the effect of different thresholds on the clustering algorithm in the range from 1.5 to 2.5. The results were added to the methods section “Robust classification of frames” as the following paragraph.

“In our analysis, we find that our modified clustering algorithm performs well for thresholds of the inconsistency coefficient ξ in the range from 1.7 to 2.2. The upper limit is determined by the minimal sensitivity needed to separate the most evident mixed-states (e.g., identified by motion-burr contrast, as in Fig. 1b). The lower limit is set by the total number of frames (here: 99%) that can be assigned to clusters that yield acceptable real-space reconstructions, which we assume as 15 frames per cluster. Specifically, we chose $\xi = 1.85$ because it produces the lowest frame misclassification probability while assigning 99.5 % of our 28.800 frames to clusters with >15 frames (see below for more details about the evaluation of the frame misclassification).”

In the second phase, we measure the misclassification error and define a threshold for how large this error may be while still claiming temporal resolution. Previously, we have just stated that we use a threshold of 1 % for this error. Now, we have modified the main text to justify this choice:

“We use this value for our further analysis because a 1 % misclassification error is low enough to allow conclusions regarding the number and frequency of reconstructed transitions.”

And finally, in the third phase, we group the internal domain modes into states according to this acceptable error threshold. The process is now explained in the Methods section *“Estimation of the temporal discrimination threshold and reconstruction of the 32 states”*:

“In Extended Data Fig. 5, we show the frame misclassification probability of all remaining mode pairs as a function of their similarity, represented by their pair correlation. Significant misclassification rates occur for very similar domain configurations only. Specifically, we find that the frame misclassification probability is below 1 % for similarities of the underlying magnetic textures of up to 93.8 %. We define this as the sensitivity threshold in our analysis. Domain modes with higher similarity were grouped allowing their temporal discrimination with an acceptable misclassification rate. These groups of modes are referred to as spatially and temporally resolved magnetic “states”. In practice, we successively combined the nearest modes until the distance between any pair of modes of two dissimilar states exceeded the 100 % - 93.8 % = 6.2 % similarity distance defined by the sensitivity threshold. An overview of the identified states and their associated internal domain modes is shown in Extended Data Fig. 6.”

In the following we list the small changes of the results between initial and revisited classification approach:

- The number of identified domain modes, decreased from initially 85 to 72 modes.
- The highest mode similarity allowing a frame misclassification probability below 1% decreased from initially 96% to 94% similarity.
- The number of states reduced from initially 36 to 32 due to the lower similarity threshold. However, states are still assigned to 99.5% of the 28800 frames.

We underline that while our modification of the algorithm has changed the number of states and internal modes, it has not changed any relevant aspects of Fig. 4 nor any of our conclusions.

2. Based on the above, the authors should be careful about some conclusions. For instance, fast oscillatory hopping between states shown in the temporal evolution in Fig. 4b may be affected by a specific choice of the above mentioned parameters.

We agree with the referee that changes of the mentioned clustering parameters, especially the misclassification threshold used to group the domain modes into the spatiotemporally resolved states, influence the temporal reconstruction. In general, the more internal modes are grouped, the less oscillations are observed.

In principle, it is indeed possible that the observation of fast oscillations and extended calm periods of the same state is related to different internal modes that we grouped into a “state”: where one mode may have a low energy barrier, and hence exhibits frequent transitions, the other may be more strongly pinned and therefore stationary.

However, this scenario is reasonably unlikely as it would mean that there are large energy barriers between different internal modes of the same state, such that intra-state transitions are much more rare than inter-state transitions. Given the very high similarity of internal modes of the same state, this scenario seems unlikely. However, we fully agree that our data is not conclusive in this regard and that our conclusions should be phrased more carefully. Specifically, the corresponding part in the main text now reads:

“Assuming that the existence of internal modes does not interfere with the concept of “lifetime” of a state as a whole, this behavior is incompatible with the established picture of Arrhenius activation over a fixed energy barrier, where longer lifetimes are expected to be exponentially suppressed. Instead, this observation suggests a dynamic energy landscape due to correlation effects of the multi-domain state.”

Note further that the observation of oscillations cannot be attributed to frame misclassification. This is guaranteed by our very low misclassification error tolerance. To support this statement, we have added in Figure 4d image reconstructions from just the frames corresponding to the shown oscillations. These reconstructions are clearly different and show no signs of mixed-state superposition, e.g., contrast degradation or motion blur, hence confirming that rapid oscillations are indeed real.

3. Challenge in application to other systems might be the scalability of the approach, i.e., number of frames required to record more than the number of states considered in the present study.

We can, of course, only observe states that existed for a minimum total number of frames. In our case, we are able to reconstruct states that existed for at least 15 frames which is less than 6s in the almost 5h observation time. If a system transitions between a larger number of states, then more data would be required to fully capture this behavior. However, this is a general truth not specific to our technique.

In fact, CCI works also if the system has not yet explored all its states. This fact, and the scalability between small and large data sets, is already included in our manuscript: while the algorithm is explained based on small data sets of only 250 frames (Figs. 3 and E3), the same algorithm applies equally to the entire set of 28,800 frames.

We note, that for an increasing amount of data, other methods used specifically in the clustering of high dimensional data might be more practical. The classification could also benefit from developments in the highly active research fields of artificial intelligence and machine learning.

4. There are some misprints in the manuscript:

- Page 20, lines 550-551, should be “(panel d)”, instead of “(panel h)”

- Pages 20-21, in Section “Robust classification schemes” one has to replace “Fig. 2” everywhere by “Fig. 3”.

We thank the referee for pointing out these mistakes, which we corrected.

Reviewer #3 (Remarks to the Author):

Rejected on grounds of specialist interest, lack of novelty, insufficient conceptual advance. The authors present as novel a technique they branded CCI Coherent Correlation Imaging, to image time dependent magnetic fluctuations.

Since the high brightness new sources became popular, Coherent Diffraction Imaging (CDI) has been at the forefront of the experiments using the coherence of X-rays for imaging. Meanwhile X-ray Photon Correlated Spectroscopy (XPCS) has been used to observe the dynamics and fluctuations in materials. Differences in the experimental set-ups, especially the detector specifications and the required coherent photon flux conditions, didn't allow the simultaneous, dual, spatial and temporal analysis of the scattered patterns recorded on those experiments. Meaning that the analysis of the XPCS data didn't allow for the reconstruction of images when performing XPCS, nor extract time dependences of the CDI images, most of it because of the need for long time exposures for good signal to noise ratios.

Scientists have been creative in how to overcome these limitations, until approximately 8 years ago, a new generation of faster area detectors provided that possibility. CSX beamline has one of these detectors. Increasing the detectors recording speed while having enough photon flux allows faster collection of data frames and make the incursion in the faster time regimes. This was mainly explored at the new emerging coherent sources, such as FELs.

We appreciate that the referee acknowledges that simultaneous reconstruction of spatial and temporal resolution is very challenging. We also agree that there is a general trend toward addressing the issue by faster detectors and higher flux sources. However, since every system is eventually modified or destroyed by a too large incident flux, its unperturbed evolution becomes ultimately fundamentally inaccessible despite these technological developments. This is in fact the case for our system, which cannot be exposed to a larger flux than used here without modifying the dynamics, even though higher flux would have been available. Therefore, in spite of the ongoing progress on x-ray sources and detectors, a wealth of phenomena will still only be accessible if these developments are complemented by new, photon-efficient methods to extract spatiotemporal information beyond the limitations of perturbative or destructive approaches.

Work has been done in the past that shows magnetic domains configurations, showing pinning points along the different magnetic states over a magnetization curve. Recently a lot of efforts has been put in showing the dynamics of skyrmions with some of the experiments been successful.

The observation of states along a magnetization curve requires an active drive: the magnetic field. Such dynamics are very different from the stimulus-free thermal hopping observed here, both in terms of their physics as well as from a technological point of view: if the dynamics is induced by an external

stimulus, then the process is no longer stochastic in time and therefore easily accessible by conventional imaging. It is the access to fully stochastic dynamics which is the key novelty of our work.

The novelty presented in this work resides mainly in the imaging reconstruction algorithm. How Fourier images are treated to finally obtain the images of magnetic domains, correlated with temporal information. The principle of the algorithm used, has been tested before with other type of samples, what is OK and provides the right background.

The recommendation is not to publish in Nature, because the interest is very specialized, not enough novelty. The referee does not denies that this could be a new tool in general to study via imaging fluctuations in materials.

However the lack of quantitative comparison, in terms of spatial-temporal resolution with other already established methods or experimental techniques, makes it difficult to foresee the improvement that CCI would bring to the field.

In our specific application we achieve sub-30 nm spatial resolution at 0.4 ms temporal resolution, which is better than the majority of magnetic imaging techniques (including all scanning techniques) but certainly not a record. However, the purpose of our work is not to present an advance of the numbers of spatiotemporal resolution, which would be rather technical, but instead to solve a fundamental dilemma, the dilemma that spatial and temporal resolution are competing. In this sense, the relevant quantitative comparison is already included in the manuscript: for the case of coherent imaging (a concept that is widely considered as the future of x-ray imaging), we can detect the state of the system with ten times better signal-to-noise ratio with CCI compared to conventional coherent imaging. This corresponds to a 100x lower photon flux per averaging period, and directly translates to the potential of realizing 100x better temporal resolution, at no cost other than post-processing computing power. At free-electron laser sources such as the European XFEL, this gain in sensitivity can even make the difference between intra-train sensitivity (i.e., microsecond temporal resolution) and the state-of-the-art inter-train sensitivity (i.e., temporal resolution of 100 ms or worse). Because CCI is a post-processing technique without additional experimental constraints, it can improve every coherent imaging experiment, in all fields of science. This is the impact of CCI.

Furthermore, it is missing for an article aspiring to present a novel technique, a quantitative evaluation of the limits of the technique in terms of coherent photon flux (max / min) requirements for the application of CCI, the proportion of partially coherent photons, and under which beam and experimental conditions it could be improved and make more general than for the case presented.

We agree with the referee that coherence is an important aspect since we need speckles for the image classification and sufficient fringe visibility for the image reconstruction. We have added to the results section (page 5, discussion of Fig. 2) a description of parameters that affect the temporal resolution of CCI, explicitly now referring to the speckle contrast:

“On the contrary, temporal resolution emerges from the precision by which each snapshot, and hence timestamp, is assigned to the correct state. The temporal resolution of CCI is hence independent of the spatial resolution, i.e., the captured q-range, but instead depends on the similarity between states and on the speckle contrast.”

The relation between coherence and speckle contrast is textbook knowledge, a review of which would be incompatible with the limitations on the total length of the manuscript. More importantly, however, almost all synchrotrons will be upgraded to the 4th generation in the next years, and the problem of limited coherence, especially across a few-micrometer holography mask, will soon be resolved, at least in the soft and tender x-ray regime.

Overall, we agree with the referee that our manuscript does not explore exhaustively the performance of the technique as a function of all relevant external parameters. However, this is also not the scope of our work. While we still continue to actively explore such limitations, we are convinced that the technique is at a stage where it should be presented to a broad community, from scientists who need spatiotemporal access to fluctuating states of matter, to a community of x-ray and data scientists who will further develop the capabilities and understanding of the technique in the years to come.

Reviewer Reports on the First Revision:

Referees' comments:

Referee #1 (Remarks to the Author):

I investigated the rebuttal letter and revised manuscript. I have found that the authors successfully address all the comments raised by Reviewer #2 and me. Revised Fig. 3 and the introduction of inconsistency coefficient ξ are useful to explain the process of the new 'coherent correlation imaging' method.

Although Reviewer #3 cast doubt on the general interest and novelty, I think that the imaging method proposed in this study will be widely used in near future.

Referee #2 (Remarks to the Author):

The revised manuscript accounts for my comments that are also satisfactorily addressed in the direct reply by the authors to my report. Therefore, I feel that the paper is now acceptable for publication in Nature.

Referee #3 (Remarks to the Author):

The referee thanks the authors for their comments and the significant changes introduced in the resubmitted text. In their comments the authors refer to their technique applied to a sample with stochastic behavior and in special to an arduous treatment of the data to retrieve information from their sample as the innovative aspect of the paper. Simplifying, this is an experiment with data collected as already has been done in other experiments on a sample with stochastic dynamics, requiring very elaborate data analysis to extract the information of the system. In summary, this is the script of an experiment with substantial efforts in the data analysis, easy to publish in specialized journals. The added/modified information does not provide further evidence to justify publishing the manuscript in Nature. Referee advice not publication in Nature.

No quantitative comparison with the best in the field techniques parameters and improvements in the resolution that has been achieved, to show the improvement in the data collection using this type of data analysis on other type of samples without stochastic dynamics. If only useful for stochastic samples, then is too specific for publication in nature. Without comparison why should other scientists go through the exercise if there are no improvements from the already available imaging techniques?

The referee's concerns regarding the spatio-temporal resolution when recording magnetic images without providing temporal resolution have not been addressed. In the interest of providing the timing of the dynamics present at the sample, it will be necessary to provide the temporal resolution of the final images, in the assignment of the timestamps and how temporal uncertainties were

considered in the data collection and in the retrieval of the “states” images. In short, answering the question, what are the temporal uncertainties of the experiment, if “states” repeats in time and what is their time frame within the instrument/data collection time resolution. The authors wrote that they could not see any cyclic dynamics, so how can they be sure that “states” repeats in time, especially when collecting with opposite polarizations?

Note 1: Uncertainties considering, if it is possible without a trigger collect and retrieve images of the same “state” within the instrumental resolution, or if a specific “state” of the sample repeats and is available to be imaged using two opposite polarizations, considering that a stack of frames with both polarizations takes 120s to be collected. The authors may recall the experiment of using a stopwatch and trying to stop it manually, without an optical or mechanical trigger, at a generic time: XX” YY’ 0.35s, several times, and the difference when trying to stop at a specific time AA” BB’ 0.35s.

Note2: Another reason to comment on the temporal resolution, regarding cyclic dynamics, it is missing if the reason for not finding them, is because they do not exist, or because they are expected to be faster than the data collection, or the temporal uncertainties have not been considered for the statement. Not seen is not proof of not existing, maybe is just a case of not enough temporal resolution in the experiment to see it.

Reply to the Referees

We would like to thank all referees for their careful review of our revised manuscript. In particular, we thank Referee #1 and #2 for recommending publication of our manuscript in Nature, and Referee #1 for explicitly pointing out the novelty of our method and the broad application spectrum expected. We also thank Referee #3 for valuable feedback that allowed us to clarify a few things that could indeed have been misleading. Below, we reply to the comments of Referee #3 point-by-point. We reprint the Referees' feedback in normal font and our reply in bold.

Reviewer #1 (Remarks to the Author):

I investigated the rebuttal letter and revised manuscript. I have found that the authors successfully address all the comments raised by Reviewer #2 and me. Revised Fig. 3 and the introduction of inconsistency coefficient ξ are useful to explain the process of the new 'coherent correlation imaging' method.

Although Reviewer #3 cast doubt on the general interest and novelty, I think that the imaging method proposed in this study will be widely used in near future.

Reviewer #2 (Remarks to the Author):

The revised manuscript accounts for my comments that are also satisfactorily addressed in the direct reply by the authors to my report. Therefore, I feel that the paper is now acceptable for publication in Nature.

Referee #3 (Remarks to the Author):

The referee thanks the authors for their comments and the significant changes introduced in the resubmitted text. In their comments the authors refer to their technique applied to a sample with stochastic behavior and in special to an arduous treatment of the data to retrieve information from their sample as the innovative aspect of the paper. Simplifying, this is an experiment with data collected as already has been done in other experiments on a sample with stochastic dynamics, requiring very elaborate data analysis to extract the information of the system. In summary, this is the script of an experiment with substantial efforts in the data analysis, easy to publish in specialized journals. The added/modified information does not provide further evidence to justify publishing the manuscript in Nature. Referee advice not publication in Nature.

We thank the Referee for the comment, but respectfully disagree with the conclusions.

The main innovation for publishing our work in Nature is that it solves a general key issue of time-resolved imaging, namely the entanglement of spatial and temporal resolution when the number of photons is limited. If a spatial resolution of many micrometers and more is sufficient, imaging with visible light or x-rays at MHz rates (or even more) is already possible today based on high-speed detectors. Vice versa, imaging with nanometer-scale resolution is also possible using soft or hard x-rays albeit with acquisition times of many seconds or more. The fundamental reason for this limitation is that a short acquisition may not carry sufficient signal to achieve the high resolution desired. While limitations of sources, detectors, and optics may be overcome in the future, the ultimate dose limit of sample destruction will remain. This general issue is, so far, only solved for strictly repeatable dynamics using pump-probe methods. We present a fundamentally new approach, unlocking access to hitherto invisible stochastic dynamics on nanometer length scales and

millisecond timescale at synchrotron-radiation sources, and microsecond or even femtosecond (when combined with pump-probe) timescales at XFELs.

For our data analysis, we use correlation and clustering methods that are well-known for decades and are frequently applied in genome research. We further developed the clustering method to comply with data sets with many thousands of samples by introducing an iteration procedure and finally reducing the size of the cluster set by another similarity analysis. The numerical procedure is fully described in our manuscript, and we openly share our code with the community. While our data analysis is certainly more elaborate than what is needed for many direct imaging methods, we do not see any computational obstacle to an immediate application of CCI considering that advanced numerical reconstruction methods are common in the coherent imaging community and researchers are very well experienced in the application of highly developed numerical tools. Moreover, even for general users, entry barriers are reasonably low since extensive support is available at synchrotron-radiation and XFEL facilities, including support on and infrastructure for advanced data analysis.

Regarding the novelty of the experiment, we indeed use a standard coherent imaging geometry as it has been used by many researchers before and as it is available at many synchrotron-radiation and XFEL endstations worldwide. We see this as a particular advantage of our method as it dispenses from any modification of these instruments to apply CCI straight away. Also, there is already widespread experience in the community on how to record such data. However, we disagree that our “data [were] collected as already has been done in other experiments”. To our knowledge, we here present the first study where coherent scattering data were consecutively recorded over a long time span in an XPCS-like manner, however, with the goal to retrieve direct real-space information. Importantly, the acquisition time (corresponding to the time resolution) of a single frame is much smaller than what would be needed to retrieve an image from a single frame using standard imaging methods.

No quantitative comparison with the best in the field techniques parameters and improvements in the resolution that has been achieved, to show the improvement in the data collection using this type of data analysis on other type of samples without stochastic dynamics. If only useful for stochastic samples, then is too specific for publication in nature. Without comparison why should other scientists go through the exercise if there are no improvements from the already available imaging techniques?

From a methodical viewpoint, one can make a division between two types of dynamics: repeatable dynamics and non-repeatable dynamics. To be observable, repeatable dynamics must be driven by an external stimulus which can be a short trigger (such as a laser pulse to induce ultrafast demagnetization) or a continuous excitation (such as a radio-frequency field to induce magnons). Today, this type of dynamics can be well studied with nanoscale imaging methods based on the repetitive pump-probe scheme. The fundamental problem of how to acquire sufficient signal also on very short timescales is simply solved by running through the same cycle of excitation and response over and over and averaging the signal from all cycles. This scheme has been successfully applied for years resulting in time-resolved studies with temporal resolution down to picoseconds (SR sources) or even femtoseconds (XFEL). In our work, we do not address this type of dynamics as the fundamental dilemma tackled by us does not exist in this case.

Instead, we address the much more general class of non-repeatable dynamics. While repeatable dynamics, in particular when considering nanometer-scale heterogeneity, are

observed only in a small part of idealized samples and experiments, stochastic processes and fluctuations are ubiquitous in nature. Research examples listed in our manuscript comprise the physics of emerging chiral and topological states, the contribution of pinning and topology in phase transitions, and the role of spin and charge order fluctuations in high-temperature superconductivity (a list undisputed by the Referee). While progress in each of these fields enabled by time-resolved imaging would be considered a breakthrough, all existing imaging methods have been mostly blind to these phenomena. We therefore strongly dispute that the application case of stochastic dynamics would be “too specific for publication in nature”. To further underline the broad application frame of CCI, and to be explicit about possible limitations, we added two notes to the manuscript:

Note further that this approach of “binning” of similar configurations into “states” makes CCI broadly applicable even to systems exhibiting non-recurring dynamics, especially when combined with a fixed and finite field of view.

[...]

For all those systems which are currently only accessible by scattering techniques to track the evolution in reciprocal space, and which return, within the duration of the experiment and at least within a limited field of view, to the same configuration or a close approximation thereof (such that “binning” into a discrete set of “states” is possible), CCI now enables such direct imaging research.

Our method does not compete with existing time-resolved imaging methods as these methods are generally only suited for repeatable nanoscale dynamics. In this sense, we developed a fundamentally new imaging method rather than just “improvements” for the scientific community. Having said that, we agree with the Referee that a direct comparison of the capabilities of CCI to existing pump-probe methods may be helpful for the reader. We now provide this comparison as Table 1 including the temporal and spatial resolution achieved in this work, the perspectives of CCI considering specifications of sources and detectors available today, as well as published state-of-the-art from mile-stone experiments in the field of coherent imaging with magnetic contrast.

The referee's concerns regarding the spatio-temporal resolution when recording magnetic images without providing temporal resolution have not been addressed. In the interest of providing the timing of the dynamics present at the sample, it will be necessary to provide the temporal resolution of the final images, in the assignment of the timestamps and how temporal uncertainties were considered in the data collection and in the retrieval of the “states” images. In short, answering the question, what are the temporal uncertainties of the experiment, if “states” repeats in time and what is their time frame within the instrument/data collection time resolution. The authors wrote that they could not see any cyclic dynamics, so how can they be sure that “states” repeats in time, especially when collecting with opposite polarizations?

The time-resolution of our experiment corresponds to the frame rate the scattering data was recorded with. We recorded a frame every 387 ms with negligible uncertainty (jitter). We now explicitly state in the Methods section:

Timestamps of the frames were recorded by the beamline controls and acquisition system, with computer precision and hence negligible temporal error compared to the exposure and readout time. That is, the temporal resolution of our experiment was 0.387 s.

Additional uncertainty arises from the accuracy in the assignment of a particular domain configuration (state) to each of these frames. As already described in detail in the manuscript and in our previous reply, we are able to assign a state to 99.5% of the frames with a confidence of 99%. We visualize this time resolution in Fig. 4d where we show the rapid oscillation of the system between two states which partly happens from frame to frame.

It is one of the key findings of our work that the domain network monitored in our experiment intermittently oscillates between states and also returns to certain states after minutes or even hours of evolution. Still – within our time frame – the system does not cycle through the same sequence of states.

We identify if frames represent the same state – even if they occur hours apart – based on their pair correlation and the resulting distance of their Pearson similarity. This procedure is described in detail in our manuscript and verified to work reliably. In particular, this procedure is independent of the x-ray polarization each frame was recorded with. Also, the image reconstruction does not require recordings of the same state at both polarization states. We added the following explanation to the Methods section:

Importantly, this approach leaves only one time-dependent signal in the reconstruction process (I_{\pm}) and, moreover, produces the same output I_{diff} for the same state regardless of whether it was recorded with positive or negative helicity light. Thus, after this step, the temporal resolution of CCI equals the temporal resolution (here: the acquisition time) of the original single frame I_{\pm} .

Note 1: Uncertainties considering, if it is possible without a trigger collect and retrieve images of the same “state” within the instrumental resolution, or if a specific “state” of the sample repeats and is available to be imaged using two opposite polarizations, considering that a stack of frames with both polarizations takes 120s to be collected. The authors may recall the experiment of using a stopwatch and trying to stop it manually, without an optical or mechanical trigger, at a generic time: XX” YY’ 0.35s, several times, and the difference when trying to stop at a specific time AA” BB’ 0.35s.

The whole series of frames was recorded via a script, and timestamps of the frames were assigned with computer precision. The instrumental uncertainty is, therefore, negligible compared to the acquisition time (see reply to the previous comment above).

A polarization change is needed neither for the assignment of a state to a frame nor for the reconstruction of an image. A state, therefore, does not have to repeat after 120 s or last for more than 120 s to be identified and imaged. Our entire analysis is based on the classification of single frames. The change of polarization from time to time is only performed to retrieve an update of the almost constant topography image which is needed to isolate the magnet scattering.

Note2: Another reason to comment on the temporal resolution, regarding cyclic dynamics, it is missing if the reason for not finding them, is because they do not exist, or because they are expected to be faster than the data collection, or the temporal uncertainties have not been considered for the statement. Not seen is not proof of not existing, maybe is just a case of not enough temporal resolution in the experiment to see it.

We agree with the Referee that we can only detect dynamics within our time resolution. If, for example, the system would oscillate on a timescale much faster than the frame rate, two or more states would overlap in a single frame. Indeed, we find such frames with mixed

states and we are able to disentangle them as we present in Fig. E4 albeit without time resolution.

Having said that, we also stress that our finding of non-cyclic dynamics is not affected by these considerations. While a better time resolution may reveal additional intermittent oscillations, the overall evolution of the system will remain non-cyclic as, obviously, the sequence of states does not repeat within our 5h-long measurement time. The system could be cyclic on a longer timescale. We now explicitly state in the manuscript:

[...] the dynamics observed is neither cyclic nor evolving, at least not at the timescales resolved here.

Reviewer Reports on the Second Revision:

Referees' comments:

Referee #3 (Remarks to the Author):

Thank you very much for the authors response.

Unfortunately, the authors still not responding to the fundamental question what is the effect on the spatio-temporal resolution of the uncertainties, and error propagation or statistical errors introduced in the temporal resolution by the application of mathematical algorithms, or uncertainties and systematic errors in the data collection.

The authors provided as temporal resolution, the instrument resolution, or the duration of recording a frame. But unfortunately this is not consistent with the need to use many single frames, and to process them together with mathematical algorithms, with different purposes, from selection of frames, to the retrieval of the images. It has to be explained why when using complicated mathematical algorithms, they do not have an effect on the spatio-temporal resolution, while averaging has an effect.

By not discussing these factors in the resolution the authors provide a biased comparison with other techniques. If the table with the comparisons will be published it has to be revised. Under others but not exclusively, seems that there is missing an error bar on the temporal resolution of the CCI, in stochastic dynamics, or it is surprising that the spatial resolution is the same, when averaging many frames of a dynamic process, (line one), as when applying the authors data algorithm.

For the same reasons as previous reports, the referee does not recommend the publication in Nature, as the provided responses don't quantify the improvement that the proposed development present against the already existing techniques.

Note that referee #1 and #2 commented on the authors' rebuttal and latest report of referee #3 above

Referee #1 (Remarks to the Author):

I investigated the rebuttal letter and revised manuscript in the 2nd round. I found that the authors successfully address all the comments raised by Reviewer #2 and me. Revised Fig. 3 and the introduction of inconsistency coefficient ξ are useful to explain the process of the new 'coherent correlation imaging' method.

Although Reviewer #3 is still raising a doubt on the general interest and novelty, I think that the

imaging method proposed in this study will be widely used in near future.

Referee #2 (only remarks to the editor):

Referee #2 argues that the authors' approach is not in direct competition with the existing techniques, as it probes stochastic dynamics, whereas other techniques mostly probe repeatable nanoscale dynamics .

Author Rebuttals to Second Revision:

Reply to the Referees

We thank all Referees for their comments and questions to our manuscript and its revisions, which helped us to improve the quality of our work. Below, we reply to the last comments of Referee #3 point-by-point. We reprint the Referees' feedback in normal font and our reply in bold.

Referee #1 (Remarks to the Author):

I investigated the rebuttal letter and revised manuscript in the 2nd round. I found that the authors successfully address all the comments raised by Reviewer #2 and me. Revised Fig. 3 and the introduction of inconsistency coefficient ξ are useful to explain the process of the new 'coherent correlation imaging' method.

Although Reviewer #3 is still raising a doubt on the general interest and novelty, I think that the imaging method proposed in this study will be widely used in near future.

Referee #2 (only remarks to the editor):

Referee #2 argues that the authors' approach is not in direct competition with the existing techniques, as it probes stochastic dynamics, whereas other techniques mostly probe repeatable nanoscale dynamics .

Referee #3 (Remarks to the Author):

Thank you very much for the authors response.

Unfortunately, the authors still not responding to the fundamental question what is the effect on the spatio-temporal resolution of the uncertainties, and error propagation or statistical errors introduced in the temporal resolution by the application of mathematical algorithms, or uncertainties and systematic errors in the data collection.

We agree with the Referee insofar that the spatio-temporal resolution is one of the key figures for a time-resolved imaging method. However, the resolution that can be achieved in an experiment heavily depends on the sample and its dynamics as well as source, optics, and detector properties, making a general claim for CCI impossible. For this reason, we very carefully analyzed the situation for our specific experiment in order to provide a transparent view on CCI and a pathway to analyze and gauge other datasets (see Tab. 1).

CCI relies on assigning images of a discrete set of states to a discrete number of frames. There are two aspects in terms of resolution and uncertainty, which we discuss for CCI in our manuscript: First, the resolution of CCI is spatially determined by the resolution of individually reconstructed images of the states, and temporally by the rate the camera frames were recorded with. Second, as the Referee points out correctly, we have to consider statistical uncertainties from the clustering. As typical for time-resolved imaging methods, temporal and spatial uncertainties cannot be treated independently from each other. An improvement in the reliability in the temporal assignment of a state to a particular frame will, in turn, lead to a reduction of the ability to discretize different spatial configurations of the system. In other words, decreasing statistical errors in the time line (see Fig. 4b in our manuscript) necessitates compromising the detection of small changes in the real-space images. In the language adopted in our work, this corresponds to summarizing more "internal modes" to a "state".

We have analyzed this statistical error in very detail in our work. Using a threshold of 99% confidence in the assignment of a particular state to a frame (i.e., to a timestamp), we are able to discern states which differ by as little as 6%.

The authors provided as temporal resolution, the instrument resolution, or the duration of recording a frame. But unfortunately this is not consistent with the need to use many single frames, and to process them together with mathematical algorithms, with different purposes, from selection of frames, to the retrieval of the images. It has to be explained why when using complicated mathematical algorithms, they do not have an effect on the spatio-temporal resolution, while averaging has an effect.

It is the key idea of CCI that classifying the recorded camera frames through a correlation analysis allows informed averaging of frames belonging to the same state, such that the averaging process no longer comprises the temporal resolution, as done in the blind time averaging. Consequently, the temporal resolution of the methods becomes equivalent to the temporal resolution of a single frame. This idea, and the limitations from the clustering, are extensively explained in the manuscript.

By not discussing these factors in the resolution the authors provide a biased comparison with other techniques. If the table with the comparisons will be published it has to be revised. Under others but not exclusively, seems that there is missing an error bar on the temporal resolution of the CCI, in stochastic dynamics, or it is surprising that the spatial resolution is the same, when averaging many frames of a dynamic process, (line one), as when applying the authors data algorithm.

Of course, it is difficult to reduce the performance of an imaging technique to just two numbers, the spatial and temporal resolution. We tried our best to provide a fair, general and easy-to-oversee compilation of methods. The Referee is right that we did not include the statistical assignment errors (as discussed above) for CCI in the table, because they cannot be mapped to a resolution. Instead, the desired (or tolerable) assignment precision needs to be defined based on the specific research question at hand, as explained in our manuscript. Also, we thank the Referee for pointing us to the ambiguity in the first line of the table. Of course, the highest resolution in standard imaging is only reached when the sample is static over the averaging time interval. We added notes to the methods with respect to both points.

For the same reasons as previous reports, the referee does not recommend the publication in Nature, as the provided responses don't quantify the improvement that the proposed development present against the already existing techniques.